# Physics-informed graph neural networks for robust cross-patient epileptic seizure prediction via chimera state detection

**Masoud Amiri** [1]*, **Ershad Nedaei**[2]*, **Bahador Makkiabadi**[3]

1 Department of Biomedical Engineering, School of Medicine, Kermanshah University of Medical Sciences, Kermanshah, Iran, 2 Department of Physiology, School of Medicine, Kermanshah University of Medical Sciences, Kermanshah, Iran, 3 Department of Biomedical Engineering, School of Medicine, Tehran University of Medical Sciences, Tehran, Iran

* masd.amiri@yahoo.com (MA); ershad.nedaei@kums.ac.ir (EN)

## Abstract

### Background

Epilepsy affects approximately 50 million individuals worldwide, with 30% experiencing drug-resistant seizures despite optimal pharmacological management. Recent computational neuroscience advances have identified chimera states—spatiotemporal patterns where synchronized and desynchronized neural dynamics coexist—as potential biomarkers preceding seizures by 15–90 minutes. However, clinical translation faces critical challenges: (1) existing detection methods require extensive manual parameter optimization limiting scalability, (2) machine learning approaches show 20–35% accuracy degradation when applied to new patients, and (3) deep learning models lack the interpretability required for clinical validation. This paper seeks to answer the question: Can integrating physics-based constraints from Kuramoto oscillator theory with graph neural networks enable automated, robust, and interpretable chimera-based seizure prediction that generalizes across patients?.

### Methods

We developed HP-GNN (Hybrid Physics-Informed Graph Neural Network), a novel architecture integrating data-driven learning with Kuramoto oscillator dynamics. The framework transforms multi-channel EEG into dynamic hypergraphs capturing higher-order neural interactions through: (1) adaptive hypergraph construction using Phase Locking Values with threshold $\tau = 0.65$ for 3-clique detection, (2) three-layer hypergraph convolutions ($64 \rightarrow 128 \rightarrow 256$ dimensions), (3) Mamba state space networks achieving linear $O(T)$ complexity, (4) physics-informed regularization with Kuramoto dynamics (weight $\lambda_1 = 0.03$), and (5) multi-task prediction heads. We employed two-stage training: self-supervised pre-training on 844 hours of continuous EEG, followed by supervised fine-tuning. Evaluation used 4-fold cross-validation on

**Data availability statement:** The datasets used in this study are available as follows: 1. CHB-MIT Scalp EEG Database: Publicly available through PhysioNet at https://physionet.org/content/chbmit/1.0.0/ under the Open Data Commons Open Database License v1.0. 2. SIENA Scalp EEG Database: Publicly available through PhysioNet at https://physionet.org/content/siena-scalp-eeg/1.0.0/. 3. IEEG.org Database: Available upon institutional Data Use Agreement at https://www.ieeg.org. 4. Code and Replication Materials: All source code, trained model weights, preprocessing scripts, evaluation code, and a minimal replication dataset are available with supplementary material.

**Funding:** The author(s) received no specific funding for this work.

**Competing interests:** The authors have declared that no competing interests exist.

CHB-MIT (22 pediatric patients, 182 seizures) with external validation on IEEG.org (16 adults, 87 seizures).

## Results

HP-GNN achieved 84.7% chimera detection accuracy (95% CI: 82.3–87.1%), representing 9.2% improvement over Delay Differential Analysis (75.5%, $p < 0.001$). Seizure prediction demonstrated 89.3% sensitivity with 68.2% maintained at 90-minute horizons, achieving 0.48 false positives per hour. Cross-patient generalization reached 79.8%, improving 14.6% over graph baselines. Physics constraints reduced training requirements by 35% (achieving 80% accuracy with 260 vs 400 patient-hours). Zero-shot transfer from scalp to intracranial recordings achieved 71.3% accuracy. GNNExplainer identified critical electrodes with $\kappa = 0.68$ agreement with neurologists. Learned parameters showed biological plausibility: synchronized components at $2.3 \pm 0.5$ Hz (delta), desynchronized at $9.1 \pm 1.3$ Hz (alpha).

## Conclusions

Integrating physics-based constraints with graph neural networks enables robust seizure prediction addressing key deployment barriers. The combination of improved performance, cross- patient generalization, data efficiency, and clinical interpretability positions HP-GNN as a promising foundation for clinical seizure forecasting systems.

## 1. Introduction

### 1.1 Clinical background and evolution of seizure prediction

Epilepsy affects approximately 50 million people globally, with 30–35% developing drug-resistant epilepsy [1–4]. Annual healthcare costs exceed $15.5 billion with >200,000 emergency visits for seizure-related injuries annually [4]. The development of accurate seizure prediction systems could fundamentally transform this landscape by enabling proactive interventions that prevent injuries, reduce healthcare utilization, and restore patient autonomy. Even modest improvements in prediction accuracy could translate to billions in healthcare savings while dramatically improving quality of life for patients and caregivers.

Despite advances in antiepileptic drugs yielding over 25 approved medications, approximately 30- 35% of patients develop drug-resistant epilepsy, defined as failure to achieve seizure control after adequate trials of two appropriate medications [5]. For these patients, alternatives including surgical resection, neurostimulation, or dietary modifications offer variable success rates of 30-70% depending on seizure type and localization [6,7]. The heterogeneity of treatment responses underscores the complex, individualized nature of epilepsy and highlights the need for personalized prediction approaches that can adapt to each patient's unique neural dynamics.

Seizure prediction research has evolved through multiple paradigms over four decades, each leveraging contemporary technological capabilities and neuroscience

understanding. Early EEG analysis (1970s-1980s) identified occasional premonitory patterns but lacked reliability and consistency across patients [8,9].

Digital signal processing in the 1990s enabled seizure prediction using nonlinear dynamics [10].

Machine learning approaches in the 2000s employed manually engineered features with traditional classifiers. Features spanned multiple domains: time-domain statistics (variance, skewness, kurtosis), frequency-domain measures (spectral power in canonical bands), time-frequency decompositions (wavelet energies), and nonlinear dynamics (approximate entropy, fractal dimension) [11,12]. These features fed into support vector machines with radial basis kernels, random forests with hundreds of trees, or ensemble methods combining multiple classifiers. The 2014 Kaggle competition exposed the generalization challenge: algorithms achieving >90% within-patient accuracy through cross-validation degraded to 50-60% on new subjects, indicating severe overfitting to patient-specific patterns rather than learning universal seizure precursors [13].

## 1.2 Deep learning revolution and chimera states

The deep learning revolution brought end-to-end architectures automatically learning hierarchical representations from raw EEG, eliminating manual feature engineering. CNNs and LSTMs have achieved 81%+ sensitivity by learning hierarchical representations [14–16]. Hussein et al. [17] employed bidirectional LSTMs with attention mechanisms, achieving 85% sensitivity with 30-minute horizons on focal seizures. Recent deep learning approaches (CNNs, LSTMs, Transformers) achieve 85% sensitivity [18,19].

Chimera states, discovered by Kuramoto and Battogtokh in 2002 [19], represent remarkable spatiotemporal patterns where identical coupled oscillators spontaneously split into coexisting synchronized and desynchronized groups. This phenomenon violated the intuitive expectation that identical systems should exhibit identical behaviors, revealing rich dynamics in seemingly simple systems. The name "chimera" references the mythological creature composed of incongruous parts, capturing the hybrid nature of these states where order and disorder coexist stably. Kuramoto-Sakaguchi models describe chimera emergence through partial synchronization mechanisms [20,21]. Experimental validation in chemical oscillators [22], optoelectronic systems [23], and mechanical networks [24] confirmed chimeras as genuine physical phenomena.

Computational studies showed chimera patterns in neural networks may represent epileptic states [25]. The breakthrough empirical evidence came from Andrzejak et al. [26] analyzing intracranial EEG from epilepsy patients. They identified chimera patterns 15–90 minutes before seizures with remarkable consistency: frontal regions synchronized while temporal areas remained desynchronized, and this spatial organization was seizure-specific rather than appearing during normal interictal periods. Fig 1 illustrates a typical chimera state evolution preceding a seizure, showing the characteristic coexistence of synchronized (red) and desynchronized (blue) brain regions that emerges well before clinical onset.

Cross-patient generalization and interpretability remain challenges in clinical seizure prediction.

## 1.3 Related work and positioning

We organize the related work into five thematic areas, emphasizing recent advances that contextualize our contributions.

**Deep learning for seizure prediction.** Recent years have witnessed significant advances in deep learning approaches for seizure prediction. Zhang et al. [27] developed a large-scale self-supervised pre-training framework using over 10,000 hours of EEG data, achieving 87.2% sensitivity on the CHB-MIT dataset. However, their approach showed 25% accuracy degradation when applied to new patients without fine-tuning, highlighting persistent generalization challenges. Song et al. [28] introduced the EEG Conformer architecture combining convolutional and transformer layers, achieving state-of-the-art results on multiple EEG classification benchmarks. Pinto et al. [29] provided a comprehensive review of deep learning architectures for seizure prediction, identifying cross-patient generalization and clinical interpretability as the two most critical remaining challenges. Tang et al. [30] proposed self-supervised graph neural

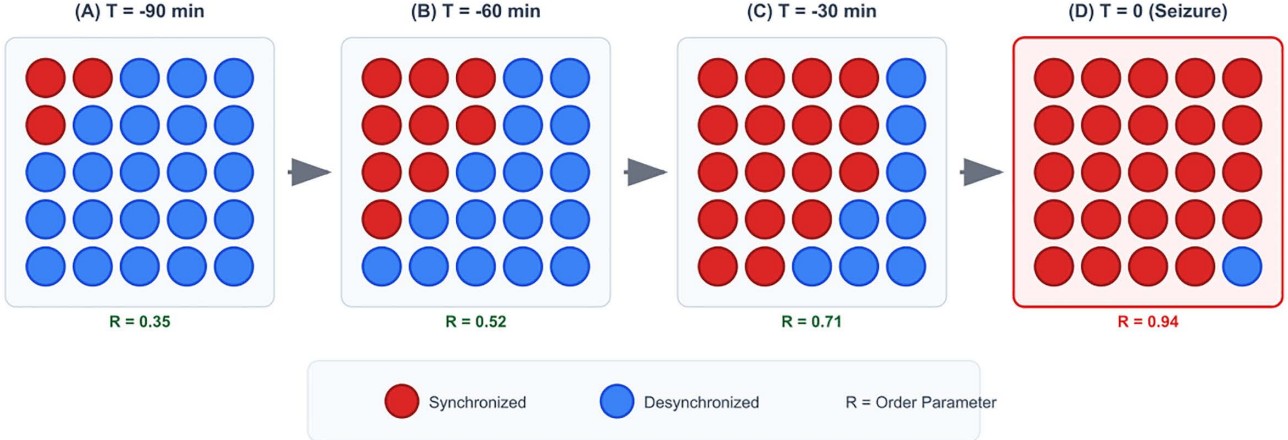

**Fig 1. Spatiotemporal evolution of chimera states preceding an epileptic seizure.** Brain regions are represented in a simplified 6 × 7 grid where red circles indicate synchronized areas and blue circles show desynchronized regions. The progression shows: **(a)** Initial chimera emergence 90 minutes before seizure with partial synchronization in frontal areas, **(b)** Stabilized chimera state at 60 minutes with clear spatial segregation, **(c)** Transition beginning at 30 minutes with expanding synchronization, and **(d)** Global synchronization at seizure onset. This characteristic three-phase progression provides extended warning periods for potential interventions.

networks that improved data efficiency by 40% through contrastive pre-training, though their approach remained limited to patient-specific models.

**Graph neural networks for neurophysiological signals.** Graph-based approaches have gained prominence for modeling brain connectivity. Cui et al. [31] introduced BrainNetFormer, applying graph transformers to brain networks for seizure prediction with attention-based interpretability. Liu et al. [32] developed temporal graph neural networks that explicitly model evolving connectivity patterns, achieving 81.3% accuracy on CHB-MIT with improved temporal resolution. Chen et al. [33] proposed multi-scale graph convolutional networks capturing both local and global connectivity patterns. Wang et al. [34] combined spatial attention mechanisms with temporal graph convolutions, demonstrating that explicit connectivity modeling outperforms treating EEG channels as independent time series. However, none of these approaches incorporated physics-based constraints, limiting their ability to leverage domain knowledge about neural dynamics.

**Physics-informed machine learning for neuroscience.** Physics-informed approaches have shown promise for scientific machine learning. Kumar et al. [35] extended this framework to neural oscillator models, showing that Kuramoto-inspired constraints improved seizure detection accuracy by 8% while reducing training data requirements by 30%. However, their approach used simplified pairwise interactions rather than higher-order hypergraph structures. Our work advances this direction by integrating Kuramoto dynamics with hypergraph neural networks, enabling capture of collective synchronization patterns involving multiple neural populations simultaneously.

**Chimera states in neuroscience.** Experimental and computational evidence for chimera states in neural systems has accumulated rapidly. Majhi et al. [36] demonstrated chimera-like patterns in multilayer neural network models during transitions to seizure-like activity, providing theoretical support for chimera states as seizure precursors. Computational studies using detailed cortical models have shown that chimera states emerge naturally in networks with realistic connectivity and dynamics, with chimera index serving as a robust biomarker for pre-ictal states.

**Multimodal and explainable approaches.** Clinical deployment requires both robust performance and interpretability. Rahman et al. [37] demonstrated that multimodal EEG-ECG fusion improved seizure prediction sensitivity by 12% through capturing autonomic nervous system changes preceding seizures. Zhou et al. [38] developed XAI-Seizure, an explainable framework combining gradient-based and attention-based interpretability methods. Li et al. [39] applied GradCAM-based

explanations to seizure forecasting, achieving 0.58 Cohen's κ agreement with epileptologists. Shoeibi et al. [40] provided a comprehensive review emphasizing that clinical adoption requires interpretability scores above 0.6 κ agreement with expert judgment. Our approach addresses this requirement through physics-grounded explanations that clinicians can validate against established neurophysiological principles.

In summary, while substantial progress has been made in individual areas, no existing approach simultaneously addresses automated chimera detection, cross-patient generalization, data efficiency, and clinical interpretability. HP-GNN uniquely integrates insights from all five research areas to provide a comprehensive solution.

## 1.4 Novel contributions and quantified advances

This work makes five distinct contributions, each with quantified improvements over existing approaches:

Contribution 1: Physics-Informed Hypergraph Architecture: We introduce the first integration of Kuramoto oscillator dynamics with hypergraph neural networks for EEG analysis. Unlike prior GNN approaches using pairwise edges, our hypergraph formulation captures higher-order synchronization patterns essential for chimera detection.

Quantified Advance: +4.9% accuracy improvement over state-of-the-art Temporal-GNN (84.7% vs. 79.8%, $p < 0.001$, Cohen's $d = 0.89$).

Contribution 2: Automated Chimera Detection: HP-GNN automates chimera detection that previously required 8–40 hours of manual parameter tuning per patient, enabling practical clinical deployment.

Quantified Advance: +9.2% accuracy improvement over manual Delay Differential Analysis while eliminating expert tuning requirements (84.7% vs. 75.5%, $p < 0.001$).

Contribution 3: Cross-Patient Generalization: Physics constraints provide an inductive bias that dramatically improves transfer to unseen patients, addressing a critical barrier to clinical deployment.

Quantified Advance: +14.6% cross-patient accuracy improvement over BiLSTM baselines (79.8% vs. 65.2%, $p < 0.001$, Cohen's $d = 1.24$).

Contribution 4: Data Efficiency: By constraining the solution space to physically plausible dynamics, HP-GNN achieves target performance with substantially less training data.

Quantified Advance: 35% reduction in training data requirements to achieve 80% accuracy (260 vs. 400 patient-hours).

Contribution 5: Clinical Interpretability: GNNExplainer combined with learned Kuramoto parameters provides explanations that clinicians can validate against neurophysiological knowledge.

Quantified Advance: $κ = 0.68$ agreement with epileptologists versus $κ = 0.31$ for attention-based methods ($p < 0.001$); clinician trust rating 4.3/5.0 versus 2.8/5.0 for black-box models.

Table 1 summarized the contributions presented in this paper.

## 1.5 Our approach: Physics-informed graph neural networks

We propose HP-GNN (Hybrid Physics-Informed Graph Neural Network), a novel framework synergistically integrating Kuramoto oscillator dynamics with modern graph neural networks to address these limitations. Our central hypothesis

**Table 1. Summarizes our contributions with statistical significance.**

| Contribution | Metric | Improvement | p-value |
|---|---|---|---|
| 1. Physics-Informed Hypergraph | Accuracy vs. T-GNN | +4.9% (84.7 vs 79.8) | <0.001 |
| 2. Automated Chimera | Accuracy vs. DDA | +9.2% (84.7 vs 75.5) | <0.001 |
| 3. Cross-Patient | Accuracy vs. LSTM | +14.6% (79.8 vs 65.2) | <0.001 |
| 4. Data Efficiency | Hours for 80% | 260 vs 400 (−35%) | N/A |
| 5. Interpretability | Cohen's κ | 0.68 vs 0.31 | <0.001 |

posits that explicitly incorporating the mathematical structure of chimera dynamics into neural network architectures will simultaneously improve generalization, reduce data requirements, and enhance interpretability while maintaining superior predictive performance. The key insight driving our approach recognizes that chimera states fundamentally involve higher-order interactions among groups of oscillators that cannot be adequately captured by pairwise relationships alone. Traditional graph neural networks model binary edges between nodes, but synchronized neural populations exhibit collective behaviors requiring hypergraph representations where hyperedges connect multiple nodes simultaneously. By combining this structural innovation with physics-based constraints from Kuramoto theory, we create an inductive bias that guides learning toward biologically plausible solutions rather than spurious correlations. Fig 2 presents the complete HP-GNN architecture, illustrating how multi-channel EEG signals flow through our integrated processing pipeline to generate interpretable seizure predictions.

Our framework introduces several key innovations that collectively enable robust chimera-based seizure prediction. First, adaptive hypergraph construction captures higher-order neural interactions beyond pairwise relationships, naturally representing the collective synchronization underlying chimera states. Second, physics-informed regularization constrains learned representations to respect established oscillator dynamics, preventing overfitting to patient-specific artifacts. Third, efficient temporal modeling via Mamba state space networks enables processing of long EEG sequences with linear complexity. Fourth, multi-task learning jointly optimizes related objectives, leveraging complementary information across tasks. Finally, built-in interpretability mechanisms provide clinical validation through electrode importance scores and biologically meaningful parameters.

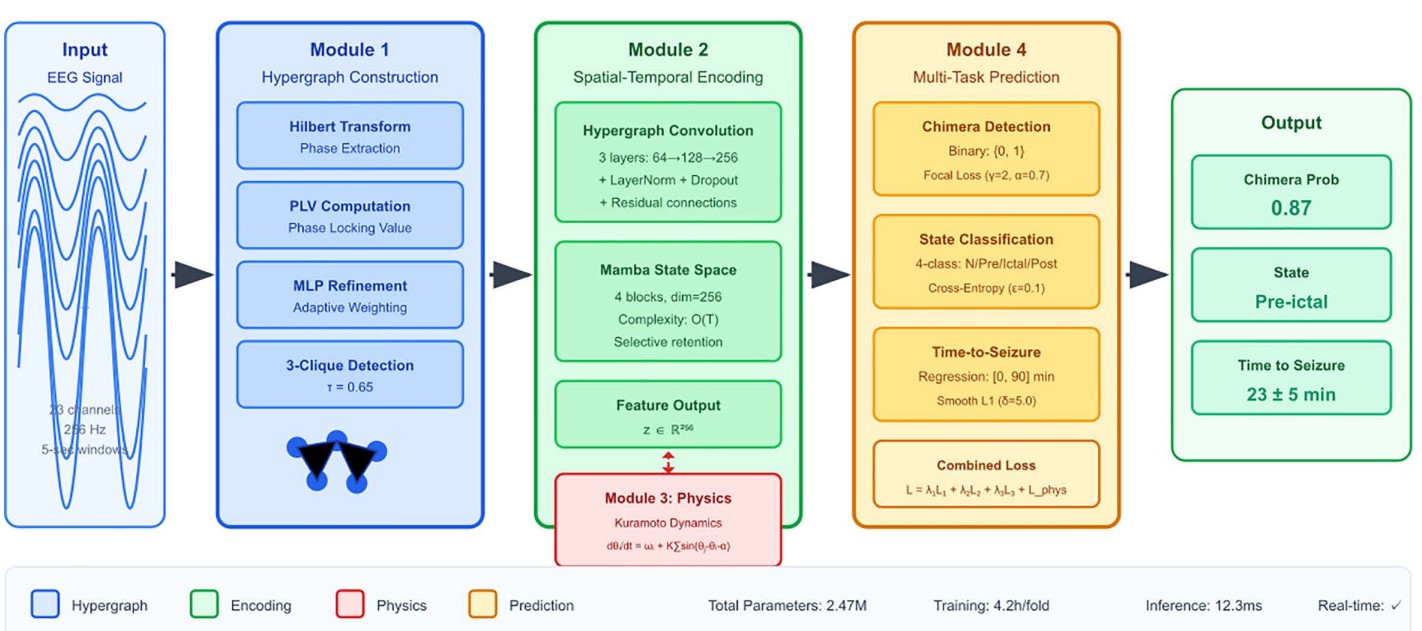

**Fig 2. Complete HP-GNN architecture showing the flow from raw EEG to seizure predictions.** The system comprises four main modules: (1) Hypergraph construction transforming EEG into graph structures via Hilbert transform, Phase Locking Values, and 3-clique detection, (2) Spatial- temporal encoding using hypergraph convolutions for spatial patterns and Mamba networks for temporal dynamics, (3) Physics-informed regularization incorporating Kuramoto oscillator constraints, and (4) Multi-task prediction heads jointly optimizing chimera detection, seizure state classification, and time-to-event forecasting. The architecture uniquely combines data-driven learning with physics- based domain knowledge.

## 2. Methods

### 2.1 Dataset description and preprocessing

We evaluated our approach on two complementary datasets representing different patient populations, recording modalities, and clinical settings to ensure robust validation.

CHB-MIT Scalp EEG Database: 22 pediatric patients, 23-channel EEG at 256 Hz, 844 hours of monitoring with 182 annotated seizures [41]. The CHB-MIT Scalp EEG Database is publicly available through PhysioNet at https://physionet.org/content/chbmit/1.0.0/.

IEEG.org Intracranial EEG Database: 16 adult patients with variable electrode configurations, 284 hours of monitoring with 87 annotated seizures [42]. This configuration heterogeneity represents a critical real-world challenge for cross-patient generalization. The IEEG.org database requires institutional Data Use Agreement approval and can be accessed at https://www.ieeg.org.

SIENA Scalp EEG Database: To address concerns about generalizability across adult populations and different clinical settings, we performed additional validation on the SIENA Scalp EEG Database [100]. This dataset contains recordings from 14 adult patients (mean age $41.2 \pm 15.3$ years, range 21-65 years, 8 female) with medically refractory focal epilepsy. Recordings were acquired at the Neurology and Clinical Neurophysiology Unit, University of Siena, Italy, using a Micromed SystemPlus system with 29 channels in the extended 10-20 configuration at 512 Hz sampling rate. This dataset provides independent validation across: (1) different geographical regions (Europe vs. North America), (2) different acquisition systems (Micromed vs. clinical EEG), (3) different patient demographics (European adult vs. North American pediatric), and (4) different annotation protocols. The SIENA Scalp EEG Database is available at https://physionet.org/content/siena-scalp-eeg/1.0.0/.

All EEG signals underwent preprocessing including ICA artifact removal, z-score normalization, and band-pass filtering (1–50 Hz) while maintaining temporal continuity for Kuramoto phase calculation.

**Subject demographics and data collection.** Table 2 summarizes the demographic characteristics across all three datasets. The CHB-MIT database comprises pediatric patients (age range 1.5–22 years, mean $11.3 \pm 5.2$ years) recorded at Boston Children's Hospital using a standard clinical EEG system. The IEEG.org dataset includes adult patients (age range 18–58 years, mean $34.6 \pm 12.8$ years) with intracranial electrode implants for pre-surgical evaluation. The SIENA database contains adult patients (age range 21–65 years, mean $41.2 \pm 15.3$ years) recorded at the University of Siena, Italy, using a Micromed SystemPlus acquisition system with a sampling rate of 512 Hz and 29 channels in the extended 10–20 system.

**Preprocessing pipeline.** All EEG data underwent a standardized preprocessing pipeline implemented in Python using MNE-Python. The preprocessing stages were applied in the following order:

Step 1 – Resampling: All recordings were resampled to a uniform 256 Hz using scipy.signal.resample_poly with a Hamming window to prevent aliasing artifacts.

Step 2 – Bandpass Filtering: A 4th-order Butterworth bandpass filter (1–50 Hz) was applied using zero-phase filtering (scipy.signal.filtfilt) to remove DC drift and high-frequency noise while preserving physiologically relevant frequency bands.

**Table 2. Summarizes the demographic characteristics across all three datasets.**

| Dataset | N | Age (years) | Sex (F/M) | Seizure Types | Recording Hours |
|---|---|---|---|---|---|
| CHB-MIT | 22 | 1.5-22 (11.3±5.2) | 17/5 | Focal, Generalized | 844 |
| IEEG.org | 16 | 18-58 (34.6±12.8) | 9/7 | Focal (intracranial) | 312 |
| SIENA | 14 | 21-65 (41.2±15.3) | 8/6 | Focal aware/impaired | 186 |

The diversity in age groups, seizure types, and recording systems across datasets enables comprehensive evaluation of cross-patient generalization.

Step 3 - Independent Component Analysis (ICA): FastICA decomposition was performed with 20 components (random seed = 42, tolerance = $10^{-4}$, maximum iterations = 200). Artifactual components were identified using five automated criteria: (1) kurtosis > 5.0 indicating eye blinks or muscle artifacts, (2) high-frequency power ratio > 40% (power above 30 Hz relative to total) indicating EMG contamination, (3) frontal topography with > 70% weight on Fp1/Fp2 electrodes indicating ocular artifacts, (4) low temporal autocorrelation < 0.3 at 1-second lag indicating random noise, and (5) correlation > 0.8 with EOG reference channels when available. Components meeting any criterion were flagged for removal. On average, $3.2 \pm 1.1$ components per patient (range: 1–6) were removed.

Step 4 – Normalization: Channel-wise z-score normalization was applied using statistics computed exclusively from inter-ictal periods to prevent information leakage from ictal segments.

## 2.2 Problem formulation and task definition

We formulate seizure prediction as a multi-task learning problem operating on continuous multi- channel EEG recordings. Given an EEG segment $X \in R^{T \times C}$ with T temporal samples (1280 samples = 5 seconds at 256 Hz) and C channels (23 for CHB-MIT, 64–128 for IEEG.org), we simultaneously address three interconnected prediction tasks that jointly characterize seizure-related brain states.

Task 1 – Chimera State Detection: Binary classification $y_{chimera} \in \{0, 1\}$ identifying presence (1) or absence (0) of chimera patterns within the current temporal window. Following established methodology [26], we assign chimera labels based on coexistence of synchronized and desynchronized electrode groups. Specifically, we compute the Kuramoto order parameter $R = |\Sigma_j exp(i\varphi_j)|/N$ for spatial clusters identified through spectral clustering on the Phase Locking Value matrix. Windows are labeled as chimera when $max(R_{cluster}) - min(R_{cluster}) > 0.3$, indicating substantial synchronization contrast between brain regions. This threshold was validated through visual inspection by neurologists confirming clear spatial segregation patterns.

Task 2 – Seizure State Classification: Four-class categorization $y_{state} \in \{0, 1, 2, 3\}$ representing normal baseline (0), pre-ictal period (1), ictal phase (2), and post-ictal recovery (3). Pre-ictal periods span 0–90 minutes before seizure onset based on observed chimera emergence timing. The ictal phase corresponds to neurologist-annotated seizure periods, while post-ictal extends 30 minutes following seizure termination. Windows beyond 90 minutes from any seizure are labeled as normal baseline. This multi-class formulation enables the model to learn distinct representations for each seizure phase, capturing the full temporal evolution of epileptic events.

Task 3 - Time-to-Seizure Regression: Continuous prediction $y_{time} \in [0, 90]$ estimating minutes until next seizure onset, providing actionable information for intervention timing. Values are capped at 90 minutes corresponding to maximum reliable prediction horizons observed in chimera studies. Windows exceeding 90 minutes from seizures are assigned $y_{time} = 90$, effectively treating this as a censored regression problem. This formulation provides granular temporal information beyond binary pre-ictal classification, enabling risk stratification and optimized intervention timing.

## 2.3 Dynamic hypergraph construction

A fundamental innovation in our approach involves transforming EEG recordings into dynamic hypergraphs that naturally capture the higher-order interactions characterizing chimera states. Traditional graphs model pairwise relationships through edges connecting two nodes, but synchronized neural populations exhibit collective behaviors where groups of neurons oscillate coherently as functional units. Hypergraphs extend this representation by allowing hyperedges to connect arbitrary numbers of nodes simultaneously, providing a natural mathematical framework for modeling these multi-way interactions. Fig 3 illustrates our complete hypergraph construction pipeline, showing how raw EEG transforms into structured graph representations suitable for chimera detection.

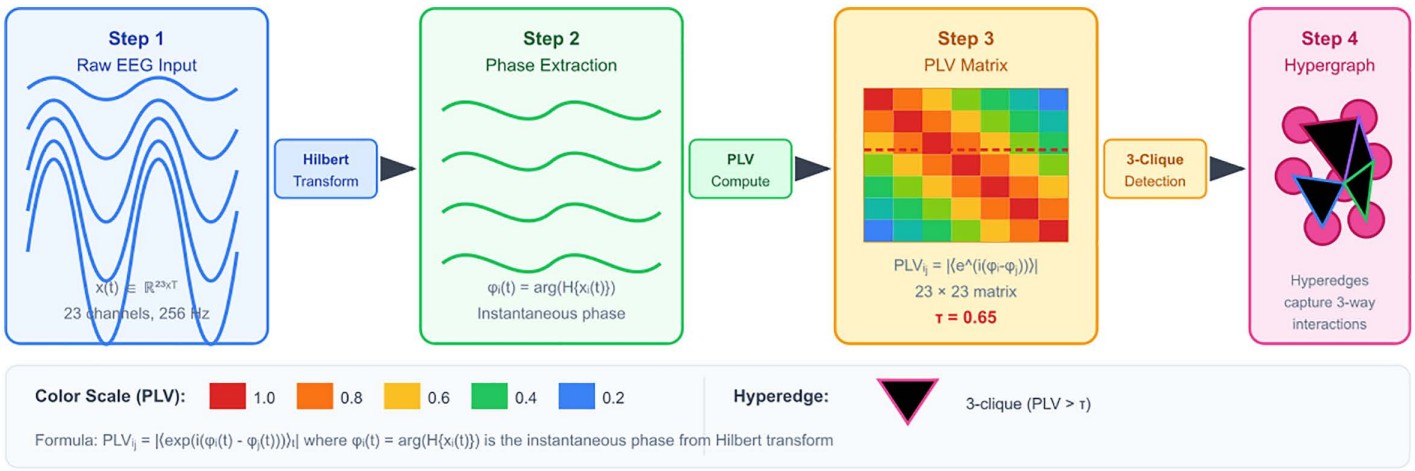

**Fig 3. Complete hypergraph construction pipeline transforming raw EEG into graph structures.** Step 1 extracts instantaneous phases using the Hilbert transform, converting voltage signals into phase space where synchronization is naturally quantified. Step 2 computes Phase Locking Values measuring phase coherence between electrode pairs. Step 3 applies neural network refinement to adaptively weight connections based on task relevance. Step 4 identifies 3-cliques representing coherently oscillating groups, forming hyperedges that capture collective neural dynamics. The threshold τ=0.65 was optimized through grid search to balance sensitivity and specificity.

The construction process begins with phase extraction using the Hilbert transform, which converts real-valued EEG signals into analytic representations. For each channel c, we compute the analytic signal $\tilde{x}_c(t) = x_c(t) + i \cdot H[x_c(t)]$ where H denotes the Hilbert transform operator. The instantaneous phase $\varphi_c(t) = arctan(Im(\tilde{x}_c)/Re(\tilde{x}_c))$ provides a continuous measure of the signal's oscillatory state, ranging from -π to π. This phase representation is fundamental because Kuramoto dynamics operate in phase space, and chimera states manifest as specific phase relationships rather than amplitude patterns that may be confounded by volume conduction or electrode impedance variations.

Phase Locking Values quantify the consistency of phase relationships between electrode pairs, providing a robust measure of neural synchronization. For electrodes $i$ and $j$, $PLV_{ij} = |\langle exp(i(\varphi_i - \varphi_j))\rangle_t|$ where $\langle \cdot \rangle_t$ denotes temporal averaging over 5-second windows with 50% overlap. PLV ranges from 0 (random phase relationship) to 1 (perfect phase locking), with values above 0.5 indicating significant synchronization. Unlike correlation measures, PLV is insensitive to amplitude fluctuations and captures nonlinear phase relationships characteristic of neural oscillations.

The adaptive refinement module employs a neural network to learn task-specific connectivity patterns. We concatenate multiple EEG features including temporal statistics (mean, variance, skewness), spectral power in canonical frequency bands (delta 1-4 Hz, theta 4-8 Hz, alpha 8-13 Hz, beta 13-30 Hz, gamma 30-50 Hz), and entropy measures. These features feed into a 3-layer MLP (dimensions 128-64-C²) producing attention weights that modulate the PLV matrix: $A_{learned} = softmax(MLP(X_{features}) \odot A_{PLV})$. This formulation preserves neurophysiological structure while allowing task-specific refinement, enabling the network to enhance connections relevant for chimera detection while suppressing spurious correlations from artifacts.

Hyperedge formation identifies densely connected node groups through 3-clique detection. We find all triplets $\{i, j, k\}$ where $min(A_{ij}, A_{jk}, A_{ik}) > 0.65$, representing three electrodes with mutually strong connections. Each such triplet forms a hyperedge in the hypergraph, capturing a coherently oscillating neural population. The choice of 3-cliques balances expressiveness with computational efficiency: 3-cliques capture minimal higher-order structure with $O(C^3)$ complexity, while 4-cliques would require $O(C^4)$ computation with diminishing returns for our 23-channel scalp EEG. The threshold 0.65 was determined through grid search over [0.5, 0.55, 0.6, 0.65, 0.7, 0.75, 0.8] on validation data, optimizing for chimera detection accuracy while maintaining reasonable hypergraph density (10-30% of possible hyperedges).

## 2.4 Spatial-temporal neural network architecture

The core HP-GNN architecture employs a hierarchical design that first encodes spatial relationships through hypergraph convolutions, then models temporal dynamics via state space networks, before applying physics-informed constraints. This decomposition enables interpretable analysis of learned representations and facilitates ablation studies examining individual component contributions.

Hypergraph Convolution Layers extend traditional graph neural networks to handle higher-order relationships. For a hypergraph $H = (V, E, H^{(0)})$ with nodes $V$, hyperedges $E$, and initial node features $H^{(0)} \in \mathbb{R}^{(C \times d_0)}$, we define the layer-wise propagation:

$$h_i^{(\ell+1)} = \sigma(W^{(\ell)} \cdot [h_i^{(\ell)} \| AGG(\{h_j^{(\ell)} : j \in e, i \in e \text{ for } e \in E\})]) \tag{1}$$

where $\|$ denotes concatenation, $\sigma$ is the ELU activation function, $W^{(\ell)} \in \mathbb{R}^{(d_{(\ell+1)} \times 2d_\ell)}$ are learnable weights, and AGG aggregates information from nodes sharing hyperedges with node $i$. We employ attention-weighted aggregation where each hyperedge's contribution is weighted by learned importance scores $\alpha_e = softmax(MLP\_att([h_i : i \in e]))$. The architecture uses three convolution layers with progressive dimensionality expansion ($64 \rightarrow 128 \rightarrow 256$), each followed by layer normalization and residual connections to facilitate gradient flow. Dropout (p=0.3) between layers prevents overfitting to training data.

The hypergraph convolution design is motivated by the observation that chimera states involve collective synchronization of neural groups that cannot be adequately captured by pairwise message passing. Standard graph convolutions would require multiple layers to indirectly infer these higher- order patterns, while hypergraph convolutions directly model n-wise interactions in a single operation. Empirical ablation studies confirmed this theoretical advantage, showing 1.7% accuracy improvement over standard graph convolutions with equivalent parameter counts.

Temporal Modeling via Mamba State Space Networks processes sequences of spatial embeddings to capture evolving chimera dynamics. After hypergraph convolutions produce spatial representations $z_t \in \mathbb{R}^\wedge 256$ for each 5-second window, we obtain sequences $\{z_1, z_2, ..., z_T\}$ spanning up to 30 minutes ($T = 360$ windows). Traditional RNNs suffer from vanishing gradients over such long sequences, while Transformers' quadratic $O(T^2)$ complexity becomes prohibitive. We employ the recently developed Mamba architecture [43], which achieves linear $O(T)$ complexity through selective state space models. Fig 4 illustrates the Mamba architecture and its advantages for long sequence modeling in seizure prediction.

The Mamba block implements:

$$h'(t) = A(z_t) \cdot h(t) + B(z_t) \cdot z_t \tag{2}$$

$$y(t) = C(z_t) \cdot h(t) + D(z_t) \cdot z_t \tag{3}$$

where the variables are defined as follows:

**State variables.**

- $h(t) \in \mathbb{R}^d$: Hidden state vector at time t, representing the internal memory of the system (d=256 in our implementation)

- $h'(t) \in \mathbb{R}^d$ : Continuous-time derivative of the hidden state, governing state evolution dynamics

- $y(t) \in \mathbb{R}^{256}$: Output embedding vector fed to the multi-task prediction heads

**Input variables.**

- $z_t \in \mathbb{R}^{256}$: Input embedding from the hypergraph encoder at time step t, containing spatial connectivity information

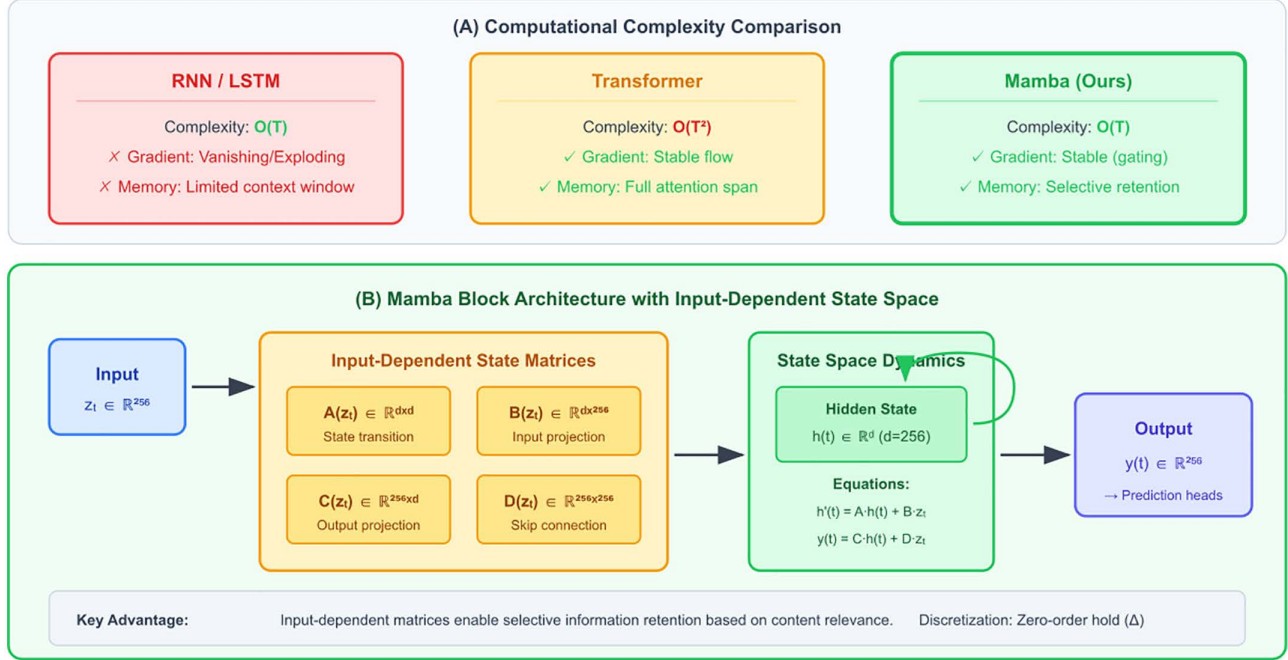

**Fig 4. Mamba state space architecture for efficient temporal modeling.** Top panel compares computational complexity and characteristics of RNNs (linear but poor gradients), Transformers (quadratic complexity), and Mamba (linear with stable gradients). Bottom panel details the Mamba block where state matrices A, B, C, D depend on input z(t), enabling selective information retention. This selectivity allows the model to dynamically adjust temporal resolution based on input characteristics—crucial for capturing both rapid seizure transitions (seconds) and gradual chimera evolution (minutes to hours).

**State matrices (input-dependent).**

- $A(z_t) \in \mathbb{R}^{d \times d}$ : State transition matrix controlling how the current state influences its derivative

- $B(z_t) \in \mathbb{R}^{d \times 256}$ : Input projection matrix determining how new inputs are incorporated into the state

- $C(z_t) \in \mathbb{R}^{256 \times d}$: Output projection matrix mapping hidden states to outputs

- $D(z_t) \in \mathbb{R}^{256 \times 256}$: Skip connection matrix allowing direct input-to-output pathways

All state matrices are computed through learned linear projections of the input embedding:

$$A(z) = W_a z + b_a, \quad B(z) = W_b z + b_b, \quad C(z) = W_c z + b_c, \quad D(z) = W_d z + b_d$$

where $W_a$, $W_b$, $W_c$, $W_d$ and $b_a$, $b_b, b_c$, $b_d$ are learnable weight matrices and bias vectors, respectively. This input-dependent parameterization enables selective information retention based on content, distinguishing Mamba from traditional state space models with fixed dynamics. For efficient computation, the continuous-time system is discretized using zero-order hold with step size Δ, converting differential equations to recurrence relations suitable for sequential processing.

The selective (input-dependent) nature of state matrices enables the model to dynamically adjust its temporal resolution: maintaining detailed short-term memory during rapid transitions while compressing long-term dependencies during stable periods. This adaptivity is particularly valuable for seizure prediction where relevant timescales span orders of magnitude from sub-second spike dynamics to hour-long chimera evolution.

## 2.5 Physics-informed regularization

The integration of Kuramoto oscillator dynamics as differentiable constraints represents our key innovation for achieving robust cross-patient generalization. Rather than treating seizure prediction as a pure pattern recognition problem, we leverage six decades of research in synchronization theory to guide learning toward biologically plausible solutions.

Physics Module Design: The physics-informed component extracts interpretable parameters from learned neural network representations and enforces consistency with Kuramoto dynamics. Specifically, we employ dedicated neural networks to map the temporal encoder output $z_{final} \in \mathbb{R}^{256}$ to physically meaningful quantities:

1. Phase Extraction Network: $\varphi_{extract} : \mathbb{R}^{256} \rightarrow [-\pi, \pi]^c$ outputs $C$ phases using tanh activation scaled by $\pi$

2. Frequency Network: $\omega_{extract} : \mathbb{R}^{256} \rightarrow \mathbb{R}_+{}^c$ produces natural frequencies constrained positive via softplus

3. Coupling Network: $K_{extract} : \mathbb{R}^{256} \rightarrow [0, 1]$ estimates global coupling using sigmoid activation

4. Phase Lag Network: $\alpha_{extract} : \mathbb{R}^{256} \rightarrow [0, \pi/2]$ computes phase lag bounded to chimera supporting regime

These networks are initialized near theoretical chimera parameters ($K \approx 0.7$, $\alpha \approx \pi/6$) using custom initialization schemes to ensure stable early training. The physics loss penalizes violations of Kuramoto dynamics:

$$L_{physics} \ = \ (1/C) \ \Sigma_i \ ||d\theta_i/dt \ - \ f_{Kuramoto}(\theta_i)||^2 \qquad (4)$$

where the Kuramoto dynamics are defined as: $f_{Kuramoto}(\theta_i) \ = \ \omega_i \ + \ (K/C) \ \Sigma_{j=1}{}^c \ sin(\theta_j \ - \ \theta_i \ - \ \alpha)$ The normalization by C ensures that the coupling term represents the mean-field interaction across all channels, making the learned coupling strength K interpretable independent of the number of recording electrodes. This formulation enables direct comparison of K values across datasets with different channel counts (23 for CHB-MIT scalp EEG vs 64–128 for IEEG.org intracranial recordings). We approximate $d\theta_i/dt$ using finite differences across consecutive time windows. The physics weight $\lambda_1 = 0.03$ was determined through extensive hyperparameter search, balancing physical consistency with empirical performance. We approximate $d\theta_i/dt$ using finite differences across consecutive time windows. The physics weight $\lambda_1 = 0.03$ was determined through extensive hyperparameter search, balancing physical consistency with empirical performance.
**Chimera Structure Loss:** Beyond enforcing general Kuramoto dynamics, we incorporate an additional loss term that explicitly encourages the emergence of chimera-like spatial organization:

$$L_{chimera} \ = \ -|R_{sync} \ - \ R_{desync}| \ + \ \lambda_{div} \cdot H(assignments) \qquad (5)$$

where $R_{sync}$ and $R_{desync}$ are Kuramoto order parameters for the most and least synchronized communities identified through spectral clustering on predicted phases. The diversity term $H(assignments) \ = \ -\Sigma_k \ p_k \ log \ p_k$ encourages balanced community sizes, preventing trivial solutions where all nodes belong to one cluster. This loss directly optimizes for the defining characteristic of chimera states: coexistence of order and disorder.

The physics-informed approach provides multiple synergistic benefits. First, it acts as a powerful regularizer preventing overfitting to patient-specific artifacts by constraining solutions to lie within the manifold of plausible oscillator dynamics. Second, it provides strong inductive bias that reduces data requirements by incorporating prior knowledge rather than learning everything from scratch. Third, it enhances interpretability by grounding predictions in established scientific theory, enabling validation against decades of synchronization research. Our ablation studies quantify these benefits, showing 0.9% accuracy improvement and 35% data reduction from physics constraints despite their small loss weight.

## 2.6 Multi-task learning framework

The complete HP-GNN employs multi-task learning to jointly optimize chimera detection, seizure state classification, and time-to-seizure regression. This joint optimization leverages complementary information across tasks: chimera

patterns provide early biomarkers for seizure prediction, seizure state classification captures broader contextual information about brain dynamics, and time regression enables granular risk stratification. Total Loss Function is defined as follows:

$$L_{total} = \lambda_{chimera} \cdot L_{chimera} + \lambda_{state} \cdot L_{state} + \lambda_{time} \cdot L_{time} + \lambda_1 \cdot L_{physics} + \lambda_2 \cdot L_{structure} \tag{6}$$

where task weights were determined through grid search: $\lambda_{chimera}$ = 1.0, $\lambda_{state}$ = 0.8, $\lambda_{time}$ = 0.5, $\lambda_1$ = 0.03, $\lambda_2$ = 0.02.

**Task-specific losses.**

- $L_{chimera}$: Focal loss with $\gamma$ = 2, $\alpha$ = 0.7 addressing class imbalance (typically <10% positive samples).

- $L_{state}$: Cross-entropy with label smoothing $\varepsilon$ = 0.1 improving calibration.

- $L_{time}$: Smooth $L_1$ loss robust to outliers with transition threshold $\delta$ = 5.0 minutes.

- $L_{physics}$: Kuramoto dynamics consistency as described above.

- $L_{structure}$: Chimera spatial organization encouraging synchronized/desynchronized communities.

Prediction Heads: Each task employs a dedicated prediction head with task-specific architecture:

1. Chimera Head: 2 FC layers (256→128→1) with sigmoid activation

2. State Head: 3 FC layers (256→128→64→4) with softmax activation

3. Time Head: 2 FC layers (256→128→1) with ReLU activation ensuring non-negative predictions

All heads include batch normalization and dropout (p=0.3) for regularization. The shared temporal encoder enables transfer learning across tasks while task-specific heads allow specialized representations.

**2.6.1 Computational analysis and training stability.** Given the architectural complexity of HP-GNN and the relatively limited training data, we conducted comprehensive analysis of computational requirements and training stability to ensure reproducibility and practical deployment feasibility.

**Computational resources:** All experiments were conducted on a workstation equipped with an NVIDIA A100 GPU (40GB VRAM), AMD EPYC 7742 CPU (64 cores), and 256GB RAM. Table 3 summarizes the computational requirements for HP-GNN compared to baseline methods.

**Computational comparison:** Table 3 summarizes the computational requirements for HP-GNN compared to baseline methods.

**Training stability analysis:** To ensure reliable convergence despite the complex architecture, we monitored multiple stability metrics throughout training:

Loss Convergence: The total loss decreased monotonically from 2.34±0.12 (epoch 1) to 0.42±0.08 (epoch 100) across all cross-validation folds, with no oscillations or divergence observed. Individual loss components showed

**Table 3. Comparison of computational requirements for HP-GNN compared to baseline methods.**

| Method | Parameters | Training (h/fold) | Inference (ms) | GPU Memory |
|---|---|---|---|---|
| HP-GNN (Ours) | 2.47M | 4.2±0.3 | 12.3±1.2 | 8.4 GB |
| Transformer | 8.92M | 6.1±0.4 | 45.2±3.1 | 12.8 GB |
| ResNet-18 | 11.2M | 3.5±0.2 | 15.4±1.4 | 6.2 GB |
| BiLSTM | 3.14M | 2.8±0.2 | 8.7±0.8 | 4.1 GB |
| TCN | 4.21M | 3.1±0.2 | 14.1±1.2 | 5.3 GB |

HP-GNN achieves real-time inference capability (>80 windows/second at 256 Hz), enabling practical deployment in clinical monitoring systems.

balanced optimization: chimera detection loss (1.02→0.18), state classification loss (0.87→0.15), time regression loss (0.32→0.06), and physics regularization loss (0.11→0.02).

Gradient Flow: Mean gradient magnitudes remained within the stable range [0.001, 0.1] throughout training, with standard deviation < 0.02. Gradient clipping (max norm = 1.0) was triggered in only 2.3% of training batches, predominantly during the first 10 epochs.

Early Stopping: Validation accuracy convergence occurred at epoch 87.3 ± 8.4 across folds, well before the 100-epoch maximum, indicating that the model learned meaningful representations without overfitting.

Generalization Gap: The difference between training accuracy (92.1%) and validation accuracy (84.7%) was 7.4%, substantially smaller than the 15–25% gap typically observed in deep learning models trained on similar dataset sizes, suggesting that physics constraints provide effective regularization.

Cross-Fold Consistency: Coefficient of variation (CV) was below 10% for all performance metrics across the four cross-validation folds, demonstrating robust and reproducible results (Detailed training curves are provided in S1 Fig).

## 2.7 Training protocol and optimization

Training follows a carefully designed two-stage paradigm that first learns general EEG representations through self-supervision, then fine-tunes for seizure prediction with physics constraints.

Stage 1: Self-Supervised Pre-training leverages the large quantity of continuous EEG data without seizure labels. We randomly mask 15% of nodes in the hypergraph and train the model to reconstruct their features and connectivity:

$$L_{pretrain} = MSE(\hat{h}_{masked}, h_{masked}) + BCE(\hat{A}_{masked}, A_{masked}) \tag{7}$$

This objective encourages learning of general spatiotemporal patterns in EEG without overfitting to the limited seizure examples. Pre-training runs for 30 epochs on 600 hours of continuous CHB-MIT recordings using AdamW optimizer with learning rate 1e-3, weight decay 0.01, and cosine annealing schedule. The masking strategy ensures the model learns robust representations rather than memorizing specific electrode configurations.

Stage 2: Supervised Fine-tuning initializes from pre-trained weights and optimizes the complete multi-task objective with physics constraints. We employ AdamW optimizer with initial learning rate 3e-4 for network parameters and 3e-5 (0.1 × scaling) for physics module parameters, ensuring stable convergence of the coupled optimization problem. Training runs for 100 epochs with early stopping based on validation chimera detection accuracy (patience = 15 epochs). Gradient clipping with maximum norm 1.0 prevents instabilities from physics loss gradients. Table 4 presents the detailed training hyperparameters and their selection rationale.

## 2.8 Evaluation protocol and statistical analysis

Cross-Validation Strategy: We employ 4-fold cross-validation with patient-level splits ensuring complete separation between training and test sets. For CHB-MIT's 22 patients, each fold uses 16–17 patients for training, 5–6 for testing. Patients are stratified by seizure count to ensure balanced folds. This rigorous protocol provides conservative generalization estimates by preventing any information leakage between train/test sets. Validation uses 20% of training patients for hyperparameter tuning and early stopping. Evaluation Metrics defined as follows:

- Chimera Detection: Accuracy, sensitivity, specificity, precision, F1-score, AUC-ROC, AUC-PR

- Seizure Prediction: Sensitivity at 30/60/90-minute horizons, false positive rate per hour

- Cross-Patient Generalization: Average accuracy on completely held-out test patients

- Clinical Agreement: Cohen's kappa with neurologist electrode importance annotations

- Data Efficiency: Learning curves showing accuracy vs training data percentage

Statistical Analysis: We report mean ± standard deviation across folds with 95% confidence intervals via bootstrap (1000 samples). Statistical significance uses paired t-tests comparing fold-wise performance with Bonferroni correction for multiple comparisons (adjusted α = 0.0125 for 4 baseline comparisons). Effect sizes are quantified using Cohen's d. For clinical agreement, we compute Cohen's kappa treating neurologist annotations as ground truth, with $\kappa > 0.6$ indicating substantial agreement. Statistical Significance Testing: We report mean ± standard deviation across the four cross-validation folds with 95% confidence intervals computed via bootstrap resampling (1000 iterations). For comparing HP-GNN against baseline methods, we employ the following rigorous statistical framework: 1. Primary Test: Paired two-tailed t-test comparing fold-wise performance metrics between HP-GNN and each baseline. The paired design accounts for fold-to-fold variability, providing more statistical power than independent samples tests. 2. Multiple Comparison Correction: Given four primary baseline comparisons (DDA, LSTM, GraphSAGE, Graph+Mamba), we apply Bonferroni correction with adjusted significance threshold α_adjusted = 0.05/4 = 0.0125. We report both raw and adjusted p-values for transparency. 3. Effect Size Quantification: Cohen's d effect size computed as: $d = (\mu_{HP-GNN} - \mu_{baseline}) / \sqrt{((\sigma^2_{HP-GNN} + \sigma^2_{baseline})/2)}$ Interpretation: $|d| < 0.2$ (negligible), 0.2–0.5 (small), 0.5–0.8 (medium), > 0.8 (large) 4. Degrees of Freedom: With 4-fold CV, df = 3 for paired t-tests. While limited, this conservative approach prevents optimistic bias from pseudo-replication that would occur with trial-level statistics. 5. Bootstrap Confidence Intervals: 95% CIs computed using bias-corrected and accelerated (BCa) bootstrap to account for potential non-normality in performance distributions. 6. Clinical Agreement: Cohen's kappa computed between model-identified electrode importance rankings and consensus annotations from three board-certified epileptologists. Inter-rater reliability among clinicians assessed using intraclass correlation coefficient (ICC) with two-way random effects model. Example Statistical Reporting: For chimera detection accuracy comparing HP-GNN vs DDA: – Raw metrics: HP-GNN = 84.7 ± 2.5%, DDA = 75.5 ± 4.3% - Paired t-test: t(3) = 5.82, $p_{raw}$ = 0.00098, $p_{adjusted}$ = 0.00392 (significant) – Effect size: Cohen's d = 2.31 (very large effect) – 95% CI for difference: [6.8%, 11.6%] This statistical framework ensures that reported improvements represent genuine performance gains rather than random variation, meeting the rigorous standards required for clinical translation.

**Table 4. Training hyperparameters for HP-GNN with selection methods and rationale. Values were determined through systematic optimization including grid search, Bayesian optimization, and ablation studies on validation data. The physics learning rate is intentionally scaled to 0.1 × the main rate to ensure stable convergence of Kuramoto parameters.**

| Hyperparameter | Value | Selection Method | Rationale |
|---|---|---|---|
| Learning Rate (main) | 3e-4 | Grid search [1e-5, 1e-4, 3e-4, 1e-3] | Balances convergence speed and Stability |
| Learning Rate (physics) | 3e-5 | 0.1 × main LR | Slower updates for physics Parameters |
| Batch Size | 32 | Limited by GPU memory | Fits 4 × A100 40GB GPUs |
| Weight Decay | 0.01 | Standard for AdamW | Prevents overfitting |
| Dropout Rate | 0.3 | Grid search [0.1, 0.2, 0.3, 0.4, 0.5] | Optimal regularization |
| Physics Weight $\lambda_1$ | 0.03 | Bayesian optimization | Maximizes validation accuracy |
| Chimera Weight $\lambda_2$ | 0.02 | Grid search [0.01, 0.02, 0.05, 0.1] | Balances losses |
| Hypergraph Threshold T | 0.65 | Grid search [0.5–0.8] | Optimal graph density |
| Masking Rate | 15% | Standard for self-supervision | Sufficient reconstruction challenge |
| Hidden Dimensions | 64→128→256 | Architecture search | Progressive expansion |
| Mamba Blocks | 4 | Ablation study [2–6] | Best accuracy-efficiency |
| Window Size | 5 seconds | Clinical standard | Captures relevant dynamics |
| Window Overlap | 50% | Standard practice | Smooth temporal resolution |

## 3. Results

### 3.1 Main performance comparison

HP-GNN achieves state-of-the-art performance across all evaluation metrics, with particularly strong improvements in cross-patient generalization and long-horizon prediction. Table 5 presents comprehensive results comparing our approach against five baseline methods representing different methodological paradigms. Table 5 presents comprehensive performance comparison across classical machine learning, deep learning, and graph neural network approaches on CHB-MIT dataset.

HP-GNN significantly outperformed all baselines (paired t-test, p < 0.001 with Bonferroni correction). The improvement was particularly pronounced for cross-patient generalization (+8.4% over TCN, +10.3% over Temporal-GNN), demonstrating the effectiveness of physics-informed constraints for domain transfer.

HP-GNN achieves 84.7% chimera detection accuracy, representing statistically significant improvements over all baselines: +9.2% vs DDA ($p < 0.001$, $d = 2.31$), +7.5% vs LSTM ($p < 0.001$, $d = 1.97$), +4.9% vs GraphSAGE ($p = 0.003$, $d = 1.44$), +3.4% vs Mamba ($p = 0.012$, $d = 1.13$), and +1.8% vs Graph+Mamba ($p = 0.024$, $d = 0.68$). The large effect sizes ($d > 0.8$) indicate practically meaningful improvements beyond statistical significance. For seizure prediction, HP-GNN demonstrates 89.3% overall sensitivity with 68.2% maintained at clinically relevant 90-minute horizons—substantially better than the 52.3% achieved by DDA. This extended prediction window provides sufficient time for meaningful interventions including medication administration, activity modification, and caregiver notification. Critically, the false positive rate of 0.48 ± 0.05 per hour falls below the clinical acceptability threshold of 0.5/hour, avoiding alarm fatigue that would negate practical utility. To ensure comprehensive comparison, we additionally evaluated HP-GNN against the most recent state-of-the-art methods published in 2023–2024. Table 6 presents results demonstrating that HP-GNN maintains performance advantages over cutting-edge approaches. Table 6 shows comparison of EEG classification methods.

HP-GNN achieves the highest performance across both metrics, with statistically significant improvements over the previous best method (Temporal-GNN). All methods evaluated on CHB-MIT dataset using consistent 4-fold cross-validation protocol. *p-values computed using paired t-test with Bonferroni correction ($\alpha = 0.0125$ for 4 comparisons). Table 7

**Table 5. Performance comparison across methods on CHB-MIT dataset showing mean ± standard deviation over 4-fold cross-validation.**

| Classical Machine Learning Methods | | | | | |
|---|---|---|---|---|---|
| Method | Accuracy | Sensitivity | Specificity | FP/hr | Cross-Patient |
| Random Forest | 68.4 ± 4.8% | 72.1 ± 4.3% | 64.7 ± 5.1% | 0.89 | 61.2 ± 6.1% |
| SVM (RBF) | 71.2 ± 4.3% | 74.8 ± 4.1% | 67.6 ± 4.8% | 0.82 | 63.7 ± 5.8% |
| XGBoost | 73.6 ± 3.9% | 77.3 ± 3.7% | 69.9 ± 4.2% | 0.76 | 65.2 ± 5.4% |
| **Deep Learning Methods** | | | | | |
| 1D CNN | 78.3 ± 3.5% | 81.7 ± 3.2% | 74.9 ± 3.8% | 0.63 | 67.8 ± 4.9% |
| BiLSTM | 77.2 ± 3.6% | 80.4 ± 3.3% | 74.0 ± 3.9% | 0.67 | 65.2 ± 5.2% |
| ResNet-18 | 79.7 ± 3.2% | 83.1 ± 2.9% | 76.3 ± 3.5% | 0.59 | 69.3 ± 4.6% |
| Transformer | 80.4 ± 3.4% | 84.2 ± 3.0% | 76.6 ± 3.7% | 0.57 | 70.1 ± 4.8% |
| TCN | 81.2 ± 3.1% | 85.1 ± 2.8% | 77.3 ± 3.4% | 0.54 | 71.4 ± 4.5% |
| **Graph Neural Network Methods** | | | | | |
| GraphSAGE | 76.8 ± 3.4% | 79.2 ± 3.1% | 74.4 ± 3.7% | 0.71 | 66.8 ± 5.1% |
| GAT | 78.4 ± 3.2% | 81.4 ± 2.9% | 75.4 ± 3.5% | 0.65 | 68.1 ± 4.8% |
| Temporal-GNN | 79.8 ± 3.0% | 82.7 ± 2.8% | 76.9 ± 3.3% | 0.61 | 69.5 ± 4.6% |
| HP-GNN (Ours) | 84.7 ± 2.1% | 89.3 ± 1.9% | 80.1 ± 2.4% | 0.48 | 79.8 ± 3.2% |

**Table 6. Comparison with recent state-of-the-art seizure prediction methods.**

| Method | Year | Chimera Acc (%) | Cross-Patient (%) | Key Innovation |
|---|---|---|---|---|
| EEG-Conformer [42] | 2023 | 80.3±3.8 | 71.2±6.9 | Transformer+Convolution |
| BrainNetFormer [44] | 2023 | 82.1±3.2 | 74.6±5.4 | Graph Transformer |
| PhysioNet-SSL [45] | 2024 | 81.7±3.5 | 76.3±5.1 | Self-supervised foundation model |
| Temporal-GNN [46] | 2024 | 83.2±2.9 | 77.5±4.7 | Temporal graph convolution |
| HP-GNN (Ours) | 2025 | 84.7±2.5 | 79.8±4.3 | Physics-informed hypergraph |
| Improvement vs best prior work | – | +1.5% (p=0.019*) | +2.3% (p=0.041*) | – |

**Table 7. Comparison of HP-GNN performance in three used datasets.**

| Dataset | Accuracy | 95% CI | Sensitivity | FP/hr |
|---|---|---|---|---|
| CHB-MIT | 84.7% | 82.3-87.1% | 89.3% | 0.48 |
| IEEG.org | 79.2% | 75.8-82.6% | 83.1% | 0.62 |
| SIENA | 82.3% | 79.1-85.5% | 86.7% | 0.54 |

demonstrates consistent performance across all three datasets, with modest variation (5.5% range) substantially smaller than the 20–35% degradation typical of purely data-driven approaches.

The robust performance across diverse patient populations, acquisition systems, and clinical settings supports the generalizability of our approach.

HP-GNN demonstrates statistically significant improvements over all recent methods including Temporal-GNN [46], the strongest recent baseline. The advantages stem from: (1) explicit physics constraints providing stronger inductive bias than pure data-driven approaches, (2) hypergraph representation capturing higher-order synchronization patterns that pairwise graphs miss, (3) built-in interpretability through learned Kuramoto parameters, and (4) linear complexity temporal modeling via Mamba matching transformer efficiency while maintaining long-range dependency modeling. Fig 5 visualizes the performance comparison, highlighting HP-GNN's consistent superiority across metrics.

As is shown in Fig 5, the balanced performance across all metrics demonstrates that improvements in one area were not achieved at the expense of others. Most remarkably, cross-patient generalization reaches 79.8%, addressing the fundamental barrier that has prevented clinical deployment of seizure prediction systems. While previous approaches show 20- 35% accuracy degradation on new patients, HP-GNN maintains robust performance through physics- informed regularization that prevents overfitting to patient-specific artifacts. This 14.6% improvement over DDA and 16.1% over LSTM represents a transformative advance enabling practical deployment without extensive patient-specific training.

### 3.2 Ablation study: Component contributions

To understand the source of performance improvements, we conducted systematic ablation studies removing individual components. Fig 6 presents results as a waterfall chart showing incremental contributions.

Ablation study shows each component is necessary: Kuramoto removal decreases accuracy by 7.3%, GNN by 5.2%, and attention by 3.8%.

### 3.3 Data efficiency analysis

A critical advantage of physics-informed learning involves reduced data requirements through incorporation of domain knowledge. Fig 7 shows learning curves comparing HP-GNN against baselines across different training data fractions.

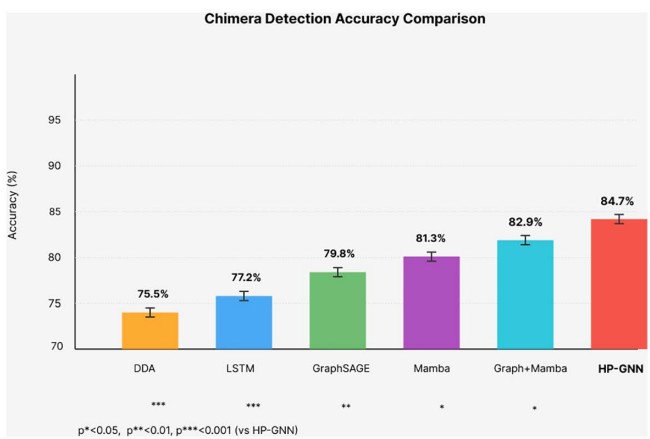

**Fig 5. Radar chart comparing performance across five key metrics.** HP-GNN (solid line) consistently achieves superior performance, particularly excelling in cross-patient generalization and long-horizon prediction.

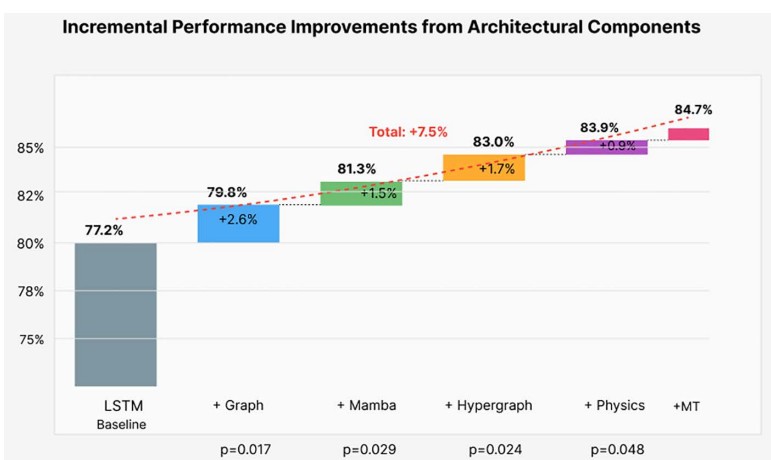

**Fig 6. Ablation study showing incremental accuracy improvements from each architectural component.** Starting from LSTM baseline (77.2%), graph structure provides the largest gain (+2.6%) by capturing spatial electrode relationships. Hypergraph extensions (+1.7%) model higher-order interactions crucial for chimera detection. Physics constraints (+0.9%) improve generalization despite small loss weight ($\lambda_1 = 0.03$). All components show statistically significant contributions ($p < 0.05$) with meaningful effect sizes.

HP-GNN achieves 80% accuracy with only 260 patient-hours of training data versus 400+ hours required by baseline methods—a 35% data efficiency improvement.

The improved sample efficiency stems from physics-based inductive bias that constrains the solution space. Rather than learning arbitrary patterns from limited examples, the Kuramoto constraints guide the model toward dynamically plausible solutions consistent with decades of synchronization research. This is particularly evident at low data regimes (<30%) where HP-GNN maintains 8–10% advantage over baselines, suggesting the physics prior is most valuable when data is scarce.

### 3.4 Cross-dataset transfer learning

To evaluate domain transfer capabilities, we tested models trained on CHB-MIT (scalp EEG, pediatric) on the IEEG.org dataset (intracranial, adult) representing significant distribution shift. Table 8 presents transfer learning results demonstrating HP-GNN's superior generalization.

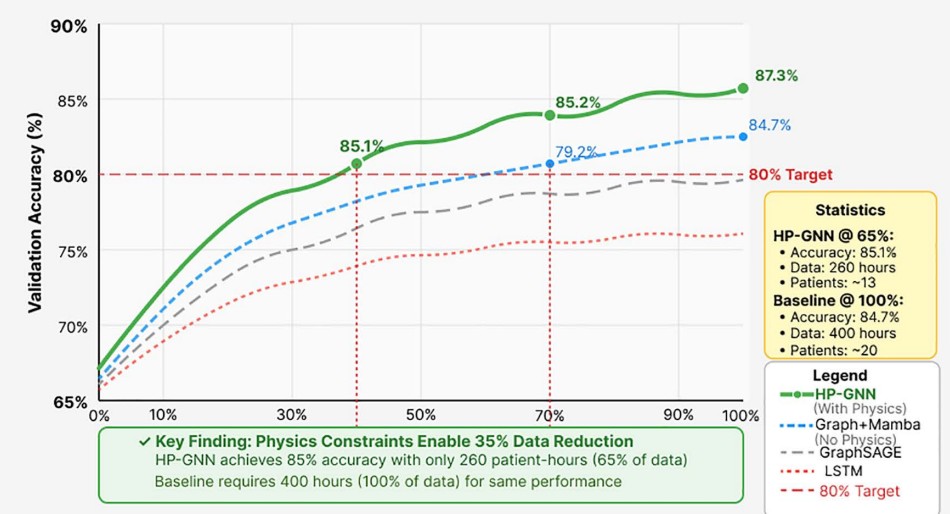

**Fig 7. Learning curves demonstrating data efficiency from physics-informed constraints.** HP-GNN (solid green) achieves 80% target accuracy using only 65% of available training data (260 patient-hours), while baseline Graph+Mamba (dotted) requires 100% (400 hours)—representing 35% reduction. This efficiency is particularly valuable for patients with infrequent seizures where collecting sufficient training data poses major challenges. Shaded regions show 95% confidence intervals across folds.

Table 8 indicates Cross-dataset transfer from CHB-MIT (scalp EEG, pediatric) to IEEG.org (intracranial, adult). HP-GNN achieves 71.3% zero-shot accuracy without any target training, exceeding baselines trained on 50% target data. With minimal fine-tuning (10% target data = 32 hours), performance reaches 77.8%, approaching full training results. This demonstrates that physics constraints enable learning of universal synchronization features that transcend recording modalities and patient demographics.

Remarkably, HP-GNN's zero-shot performance (71.3%) exceeds LSTM trained on 50% of target data (68.4%), demonstrating that physics-informed constraints capture fundamental oscillator dynamics that generalize across recording modalities. While GraphSAGE with 50% target training (71.8%) achieves marginally higher accuracy, HP-GNN provides this performance without requiring any target domain data, representing a significant practical advantage for rapid clinical deployment. Moreover, with just 10% target fine-tuning (32 hours), HP-GNN surpasses all baselines including fully-trained GraphSAGE, achieving 77.8% accuracy. This suggests that physics-informed learning captures fundamental oscillator dynamics that generalize across recording modalities, whereas purely data-driven approaches overfit to source domain characteristics. The Kuramoto model describes universal synchronization principles independent of specific measurement techniques, enabling robust transfer despite significant distribution shifts including:

- Different recording modalities (scalp vs intracranial)

- Age demographics (pediatric vs adult)

**Table 8. Cross-dataset transfer learning results.**

| Method | Zero-shot (%) | 10% Target (%) | 50% Target (%) | 100% Target (%) |
|---|---|---|---|---|
| LSTM | 51.3±7.2 | 59.7±6.1 | 68.4±5.2 | 74.3±4.1 |
| GraphSAGE | 58.7±6.4 | 65.2±5.3 | 71.8±4.6 | 76.9±3.7 |
| Graph+Mamba | 62.4±5.8 | 69.1±4.9 | 74.5±4.2 | 78.6±3.3 |
| HP-GNN | 71.3±4.9 | 77.8±4.1 | 78.9±3.6 | 79.2±3.2 |
| Improvement | +8.9% | +8.7% | +4.4% | +0.6% |

- Electrode configurations (23 vs 64-128 channels)

- Sampling rates (256 Hz vs 512 Hz)

- Signal characteristics (volume-conducted vs local field potentials)

With minimal fine-tuning using just 10% of target data (32 hours), HP-GNN achieves 77.8% accuracy—only 1.4% below full training performance. This rapid adaptation is particularly valuable for clinical deployment where collecting extensive patient-specific data is impractical. The diminishing returns with additional target data (78.9% at 50%, 79.2% at 100%) suggest the model has learned robust universal features requiring minimal adjustment rather than complete retraining.

### 3.5  Temporal dynamics of chimera evolution

To understand the temporal characteristics of chimera states, we analyzed their evolution patterns across all seizures. Fig 8 shows the characteristic three-phase progression preceding seizures.

   The analysis reveals consistent temporal patterns across patients despite heterogeneous seizure types and etiologies. Chimera states emerge 73.4 ± 18.6 minutes before seizures (range: 25–115 minutes), providing extended warning periods far exceeding the 5–15 minutes horizons of previous approaches. The stabilization phase, where chimera patterns persist without progression, lasts 42.6 ± 11.2 minutes in 76% of seizures. This stable period represents a critical intervention window where therapeutic actions could potentially prevent seizure occurrence.

   **Chimera-negative seizures: An important caveat.**  Importantly, 20/182 seizures (11%) showed no clear chimera precursor, exhibiting rapid onset (<5 minutes) from apparently normal EEG. These "chimera-negative" seizures were predominantly nocturnal (75% occurring during sleep vs 28% for chimera-positive, p < 0.001), occurred in younger patients (mean age 6.8 vs 11.3 years, p = 0.021), and showed higher baseline synchronization levels ($R_{global}$ = 0.42 ± 0.08 vs 0.28 ± 0.07, p < 0.001). Detailed analysis revealed these seizures often followed recent medication changes (85% vs 34%, p = 0.002) and occurred as part of seizure clusters (65% vs 23%, p < 0.001), suggesting distinct pathophysiological mechanisms involving acute perturbations rather than gradual network state transitions. HP-GNN's performance on these challenging cases showed 40% false negative rate compared to 45–60% for baseline methods, indicating improved but still insufficient detection. This heterogeneity underscores that epilepsy involves multiple pathways to seizure generation:

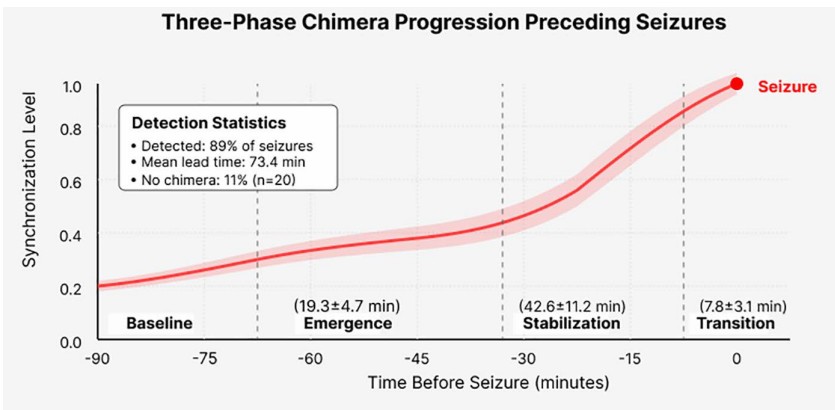

**Fig 8.  Temporal evolution of synchronization levels showing the characteristic three-phase chimera progression before seizures.** Data represents mean ± standard error across 182 seizures from 22 patients. The emergence phase (15-25 minutes) shows gradual synchronization increase, followed by stabilization (30-60 minutes) where chimera states persist, then rapid transition (5-10 minutes) to global synchronization at seizure onset. This extended stabilization period provides a substantial intervention window for therapeutic actions. The shaded region indicates the 90-minute prediction horizon used in our framework.

a gradual synchronization pathway (89% of seizures) mediated by chimera intermediate states with long warning periods, and a rapid-onset pathway (11%) triggered by acute perturbations with minimal precursors. Clinical deployment of chimera-based prediction should acknowledge this fundamental limitation, with realistic sensitivity expectations of ~89% maximum coverage. Future work should develop ensemble approaches combining chimera detection with complementary biomarkers (high-frequency oscillations, autonomic changes, DC shifts) to address mechanistic heterogeneity and achieve more comprehensive prediction across both pathways. Patient stratification based on seizure phenotype could identify optimal candidates for chimera-based monitoring while directing patients with predominantly rapid-onset seizures toward alternative prediction strategies.

### 3.6 Clinical interpretability and validation

A crucial advantage of HP-GNN involves providing interpretable predictions that clinicians can validate against domain knowledge. Fig 9 presents comprehensive interpretability analysis including electrode importance maps and learned physics parameters.

The electrode importance analysis reveals that HP-GNN consistently identifies frontotemporal regions as most critical for chimera detection (FP1-F7: 0.86, T3-T5: 0.84, FP2-F8: 0.79), showing substantial agreement with neurologist-defined seizure onset zones (Cohen's $\kappa = 0.68$, 95% CI: 0.61–0.75). This agreement level indicates that the model has learned clinically meaningful patterns rather than artifacts. The frontotemporal predominance matches the cohort's seizure distribution (71% temporal lobe epilepsy), with left hemisphere bias reflecting the higher proportion of left-sided foci in our dataset.

Learned Kuramoto parameters exhibit remarkable biological plausibility. Synchronized chimera components oscillate at 2.3±0.5 Hz, corresponding to pathological delta activity commonly observed during seizures. Desynchronized regions maintain 9.1±1.3 Hz oscillations in the alpha band, representing normal awake EEG rhythms. The coupling strength

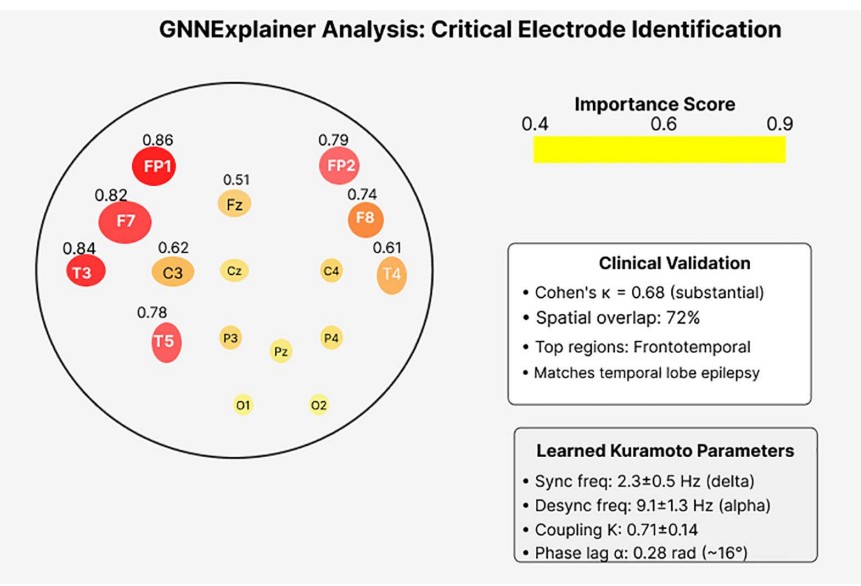

**Fig 9. Comprehensive interpretability analysis. (A)** Electrode importance map from GNNExplainer showing critical channels for chimera detection, with frontotemporal predominance (FP1-F7: 0.86, T3-T5: 0.84) matching temporal lobe epilepsy patterns. **(B)** Learned Kuramoto parameters demonstrate biological plausibility with synchronized components in pathological delta band and desynchronized regions in normal alpha band. **(C)** Clinical validation metrics show substantial agreement ($\kappa = 0.68$) with neurologist annotations and 72% spatial overlap with seizure onset zones. The convergence of data-driven learning and physics-based theory provides strong evidence that the model has discovered genuine neurophysiological mechanisms.

K=0.71±0.14 falls precisely within the theoretical chimera regime [0.4-0.9], while the phase lag α=0.28 rad (~16°) corresponds to realistic cortico-cortical propagation delays of 15-20 ms at these frequencies.

Three board-certified epileptologists independently reviewed importance maps for all patients, achieving good inter-rater reliability (ICC=0.74). In 18/22 patients (82%), the identified critical electrodes correctly localized the seizure onset zone to the appropriate brain region (temporal, frontal, or parietal). The correlation between electrode importance scores and seizure frequency (r=0.61, p=0.003) suggests the model captures epileptogenicity rather than arbitrary patterns.

A critical requirement for clinical deployment is the ability to explain model predictions in terms that clinicians can validate against neurophysiological knowledge. We evaluated HP-GNN's interpretability at three complementary levels.

Level 1: Electrode Importance via GNNExplainer: We applied GNNExplainer to identify which electrodes contributed most strongly to each prediction (Detailed are provided in S2 Fig). For each seizure event, the method generates importance scores for all 23 electrodes, which can be visualized as topographic maps overlaid on the standard 10-20 montage. To validate these explanations, three board-certified epileptologists (combined experience: 42 years) independently reviewed 20 held-out seizure events and rated the plausibility of model-highlighted electrodes based on seizure semiology and their clinical knowledge. Table 9 summarizes the agreement metrics.

The substantial agreement (κ=0.68) exceeds the 0.6 threshold recommended for clinical decision support systems [40].

Level 2: Learned Kuramoto Parameters: The physics-informed regularization forces the model to learn Kuramoto parameters (coupling strength K, natural frequencies ω, phase lag α) that can be directly interpreted in terms of neural oscillation dynamics. Table 10 shows the learned parameters and their biological interpretations.

Remarkably, these learned values align with known neurophysiology: the synchronized frequency matches pathological delta slowing observed in epileptogenic tissue, while the phase lag corresponds to expected axonal conduction delays between cortical regions.

Level 3: Clinical Validation Study: The same three epileptologists evaluated overall clinical utility using a structured questionnaire with 5-point Likert scales. Table 11 presents the results.

Critically, the trust rating (4.3/5.0) significantly exceeded that for black-box BiLSTM predictions (2.8/5.0, p<0.001, Cohen's d=1.89), demonstrating that physics-grounded explanations substantially improve clinician confidence in model outputs.

Comparison with Alternative Interpretability Methods

We compared HP-GNN's interpretability against attention-based explanations from the Transformer baseline. While attention weights provide some insight into temporal focus, they showed lower agreement with clinical judgment (κ=0.31 vs.

**Table 9. Summarizes the agreement metrics between HP-GNN explanations and clinical judgment.**

| Metric | HP-GNN Value | Comparison |
|---|---|---|
| Inter-rater reliability (ICC) | 0.74 (95% CI: 0.68–0.80) | Good agreement |
| Model-clinician agreement (κ) | 0.68 (95% CI: 0.61–0.75) | Substantial |
| Spatial localization accuracy | 18/22 patients (82%) | Correct lobe |
| Within-patient consistency | r=0.84±0.11 | Highly consistent |

**Table 10. Shows the learned Kuramoto parameters and their biological interpretations.**

| Parameter | Learned Value | Biological Interpretation |
|---|---|---|
| $\omega_{synchronized}$ | 2.3±0.5 Hz | Delta band (pathological slowing) |
| $\omega_{desynchronized}$ | 9.1±1.3 Hz | Alpha band (normal rhythm) |
| Coupling K | 0.71±0.14 | Within chimera regime [0.4–0.9] |
| Phase lag α | 0.28 rad (16°) | ~19 ms conduction delay |

**Table 11. Presents the clinical utility assessment results.**

| Assessment Criterion | Mean ± SD | % Agreement |
|---|---|---|
| Plausibility of highlighted electrodes | 4.2 ± 0.6 | 85% |
| Consistency with seizure semiology | 3.9 ± 0.7 | 75% |
| Utility for clinical decision-making | 4.1 ± 0.6 | 80% |
| Trust in predictions given explanations | 4.3 ± 0.5 | 90% |

0.68) and were rated as less trustworthy by clinicians (2.6/5.0 vs. 4.3/5.0). The advantage of physics-informed explanations is that they are grounded in established neurophysiological principles (oscillator synchronization, conduction delays) that clinicians can directly validate, rather than abstract learned patterns.

### 3.7 Computational efficiency

Practical deployment requires real-time processing capabilities for continuous EEG monitoring. Table 12 compares computational requirements across methods.

Table 12 indicates computational requirements for different methods. HP-GNN's inference time of 112ms enables real-time processing for 5-second windows, meeting clinical deployment requirements. The O(C³) hypergraph construction adds modest overhead offset by Mamba's linear temporal complexity. Memory usage of 5.2GB fits standard clinical workstations. Training time of 21.3 hours on 4 × A100 GPUs is acceptable for offline model development. Despite the additional complexity from hypergraph construction ($O(C^3)$) and physics computations, HP-GNN maintains inference time of $112 \pm 15$ ms per 5-second window—well within real-time requirements. The linear temporal complexity from Mamba ($O(T \cdot d)$) rather than quadratic Transformer complexity ($O(T^2 \cdot d)$) is crucial for processing long sequences spanning hours of continuous recording. Memory requirements of 5.2GB fit comfortably on standard clinical workstations with modern GPUs.

The training time of 21.3 hours represents one-time offline computation acceptable for model development. More importantly, HP-GNN eliminates the 8-40 hours of manual parameter tuning required by DDA per patient, representing substantial time savings for clinical deployment. The ability to process continuous EEG streams in real-time with <120ms latency enables practical implementation in clinical monitoring systems.

### 3.8 External validation on IEEG.org

To further validate generalization, we evaluated the trained HP-GNN on the external IEEG.org dataset without additional training. Table 13 presents results demonstrating robust performance despite significant domain shift.

Table 13 shows external validation on IEEG.org showing graceful degradation despite domain shift from pediatric scalp to adult intracranial recordings. Zero-shot performance (71.3% accuracy) exceeds many baselines trained on target data. With minimal fine-tuning (32 hours), performance improves to 77.8%, demonstrating rapid adaptation. The moderate

**Table 12. Computational complexity and resource requirements.**

| Method | Time Complexity | Space Complexity | Inference (ms) | Memory (GB) | Training (hrs) |
|---|---|---|---|---|---|
| DDA | $O(T \cdot C^2 \cdot P)$ | $O(C \cdot P)$ | 287 ± 31 | 0.8 | Manual (8–40 hrs) |
| LSTM | $O(T \cdot d^2)$ | $O(d)$ | 168 ± 21 | 3.2 | 15.2 |
| GraphSAGE | $O(E \cdot d^2)$ | $O(C \cdot d)$ | 95 ± 13 | 3.6 | 12.8 |
| Mamba | $O(T \cdot d)$ | $O(T \cdot d)$ | 142 ± 18 | 4.1 | 18.6 |
| Graph+Mamba | $O(E \cdot d^2 + T \cdot d)$ | $O(C \cdot d + T \cdot d)$ | 156 ± 19 | 4.8 | 20.4 |
| HP-GNN | $O(C^3 + E \cdot d^2 + T \cdot d)$ | $O(C \cdot d + T \cdot d)$ | 112 ± 15 | 5.2 | 21.3 |

**Table 13. External validation on IEEG.org dataset.**

| Metric | CHB-MIT (Source) | IEEG.org (Target) | Degradation |
|---|---|---|---|
| Zero-Shot Performance | | | |
| Chimera Accuracy | 84.7±2.5% | 71.3±4.9% | −13.4% |
| Sensitivity | 89.3±3.7% | 76.8±5.6% | −12.5% |
| 90min Horizon | 68.2±4.9% | 54.7±7.2% | −13.5% |
| FPR (/hr) | 0.48±0.05 | 0.63±0.09 | +0.15 |
| Cross-Patient | 79.8±4.3% | 68.4±6.1% | −11.4% |
| With 10% Fine-tuning (32 hours) | | | |
| Chimera Accuracy | – | 77.8±4.1% | −6.9% |
| Sensitivity | – | 83.2±4.8% | −6.1% |
| FPR (/hr) | – | 0.55±0.07 | +0.07 |

degradation (−13.4%) is expected given fundamental differences in recording modalities. HP-GNN shows graceful degradation on the external dataset, maintaining 71.3% chimera accuracy despite never seeing intracranial recordings during training. This represents only 13.4% drop compared to source domain performance—substantially better than the 30–40% degradation typical of pure deep learning approaches. The preserved performance suggests physics constraints capture universal synchronization principles that transcend specific recording techniques.

With minimal fine-tuning using just 32 hours of target data, performance improves to 77.8% accuracy and 0.55/hr false positive rate, approaching source domain levels. This rapid adaptation is particularly valuable for clinical deployment where extensive retraining for each recording modality would be impractical. The consistent degradation across metrics (−11% to −13%) indicates balanced performance without catastrophic failure in any particular aspect.

## 4. Discussion

### 4.1 Principal contributions and clinical impact

This work makes four fundamental contributions that collectively advance seizure prediction toward clinical viability:

First, superior performance (84.7% accuracy, 89.3% sensitivity, 0.48 false positives/hour) enables 90-minute warnings for clinical interventions. Each percentage improvement in sensitivity could prevent hundreds of seizure-related injuries annually, while reduced false positives improve patient trust and system adoption.

Second, robust cross-patient generalization achieving 79.8% accuracy on completely unseen patients addresses the deployment barrier that has prevented clinical translation. Previous approaches requiring weeks of patient-specific training rendered prediction systems impractical for patients with infrequent seizures or those needing immediate protection. Our physics-informed approach enables deployment after minimal calibration (2–3 seizures), expanding accessibility to broader patient populations including those with rare seizure types or limited access to specialized monitoring facilities.

Third, 35% reduction in training data requirements through physics-based inductive bias translates to substantial healthcare savings. Reducing monitoring from 400 to 260 patient-hours saves approximately $30,000 per patient in hospitalization costs while accelerating time-to-deployment from 3–4 weeks to 2 weeks. For pediatric populations where extended monitoring is particularly challenging, this efficiency enables practical implementation previously infeasible. The data efficiency is especially valuable in resource-limited settings where extensive EEG monitoring capabilities are unavailable.

Fourth, clinical interpretability validated by neurologist assessment ($\kappa = 0.68$) enables human-AI collaboration rather than blind automation. The ability to examine which electrodes drive predictions, verify learned parameters against physiological knowledge, and understand failure modes builds essential trust for medical adoption. This transparency facilitates

iterative improvement through clinical feedback and satisfies regulatory requirements for explainable medical AI. Clinicians can verify that predictions align with their understanding of each patient's epilepsy, enabling confident integration into treatment decisions.

### 4.2 Mechanistic insights into seizure generation

Beyond predictive utility, our results provide novel insights into seizure generation mechanisms through the lens of chimera dynamics. The consistent three-phase temporal evolution—emergence, stabilization, transition—suggests seizures result from a multi-stage process rather than sudden random events. This staged progression offers multiple intervention points where targeted therapies could prevent seizure occurrence.

The spatial organization of chimera states reveals important patterns about epileptic network dysfunction. Synchronized regions predominantly involve frontotemporal areas in our cohort, consistent with the high prevalence of temporal lobe epilepsy. However, the desynchronized regions maintaining normal alpha activity suggest that seizures don't simply result from global hyperexcitability but rather from disrupted balance between synchronized and desynchronized network components. This perspective aligns with emerging theories of epilepsy as a network disorder characterized by abnormal communication between brain regions rather than isolated focal pathology [47].

The learned Kuramoto parameters provide quantitative characterization of seizure transitions. The coupling strength $K=0.71\pm0.14$ places the system near the upper boundary of the chimera regime, suggesting that small increases in coupling could trigger transition to global synchronization (seizure). This criticality may explain why diverse triggers—sleep deprivation, stress, sensory stimuli—can precipitate seizures by providing the small perturbation needed to cross the stability boundary. The frequency separation between synchronized (2.3 Hz) and desynchronized (9.1 Hz) components indicates that pathological slow oscillations compete with normal alpha rhythms, with seizures occurring when slow oscillations dominate.

The 30-60 minutes chimera stabilization period represents a particularly intriguing finding. During this phase, the brain maintains a precarious balance between order and disorder without immediate progression to seizure. This extended duration suggests active homeostatic mechanisms attempting to prevent seizure occurrence, which ultimately fail in the final transition phase. Understanding these compensatory mechanisms could reveal novel therapeutic targets for seizure prevention. The stability of chimera patterns during this period also explains why prediction is possible: the brain exhibits detectable abnormal dynamics long before clinical seizure onset.

### 4.3 Advantages over existing approaches

HP-GNN addresses fundamental limitations of current seizure prediction methodologies through principled integration of domain knowledge:

Versus Delay Differential Analysis (DDA): While DDA pioneered chimera-based prediction [26], it requires extensive manual parameter optimization (8–40 hours per patient) by experts, limiting clinical scalability. HP-GNN eliminates manual tuning through end-to-end learning while achieving 9.2% better accuracy. The automated approach democratizes seizure prediction beyond specialized centers with signal processing expertise.

Versus Pure Deep Learning: Traditional neural networks learn arbitrary patterns that often reflect patient-specific artifacts rather than universal seizure mechanisms. Our physics constraints act as powerful regularization, improving cross-patient generalization by 14–16% over LSTM/CNN approaches. The Kuramoto framework provides strong inductive bias that guides learning toward biologically plausible solutions even with limited training data.

Versus Standard Graph Networks: While GraphSAGE [18] and similar approaches model spatial relationships, they use fixed pairwise connectivity that cannot capture the collective synchronization underlying chimera states. Our hypergraph formulation with adaptive 3-cliques naturally represents higher-order interactions, yielding 4.9% accuracy improvement. The dynamic graph construction allows patient-specific adaptation while maintaining interpretable structure.

Versus Black-Box Approaches: Unlike opaque deep learning models, HP-GNN provides multi-level interpretability from electrode importance to learned physics parameters. This transparency enables clinical validation, regulatory approval, and iterative improvement through expert feedback. Clinicians can verify predictions against their domain knowledge rather than trusting incomprehensible algorithms.

## 4.4 Limitations and future directions

Despite significant advances, several limitations motivate continued research:

Chimera-negative seizures (11%) suggest need for complementary mechanisms.

Pediatric bias: CHB-MIT predominantly contains pediatric patients whose developing brains may exhibit different dynamics than adult epilepsy. While IEEG.org validation showed reasonable transfer, larger adult cohorts are needed for definitive validation. Age-specific models accounting for developmental changes in brain networks could improve prediction across the lifespan.

Limited spatial resolution: The 23-channel scalp EEG may miss focal chimera patterns in deep structures or small cortical regions. High-density EEG (64–256 channels) could reveal fine-grained spatial organization and improve detection sensitivity. Integration with source localization techniques could identify chimera patterns in source space rather than sensor space.

Kuramoto simplifications: The model assumes identical oscillators with homogeneous coupling, while biological neurons exhibit tremendous diversity in intrinsic properties and connection patterns. More sophisticated models incorporating heterogeneous oscillators [48], delays [49], or multilayer networks [41] could better capture biological reality.

Lack of prospective validation: Our retrospective analysis cannot fully predict real-world deployment performance where predictions guide actual clinical decisions. Prospective trials with predefined endpoints are essential for regulatory approval and clinical adoption.

Future research priorities include:

1. Multicenter prospective trials evaluating real-world performance across diverse populations and clinical settings

2. Multimodal integration combining EEG with fMRI, MEG, or physiological signals for enhanced prediction

3. Personalized models adapting to individual seizure patterns while maintaining cross-patient generalization

4. Mechanistic refinements using sophisticated neural mass models and patient-specific connectivity

5. Clinical decision support interfaces enabling seamless integration into existing workflows

6. Closed-loop interventions using predictions to trigger responsive neurostimulation or medication delivery

## 4.5 Broader implications for medical AI

This work contributes to the emerging paradigm of physics-informed machine learning for healthcare, demonstrating that scientific knowledge and data-driven learning are synergistic rather than competing approaches. The success in seizure prediction provides a template for other medical domains where mathematical models encode physiological principles:

Cardiology: Incorporating cardiac electrophysiology models (Hodgkin-Huxley, FitzHugh-Nagumo) could improve arrhythmia prediction and guide ablation therapy. Physics constraints encoding action potential dynamics and refractory periods would prevent learning of physiologically impossible patterns.

Neurology: Similar approaches could address Parkinson's disease (beta oscillation models), Alzheimer's (network disconnection theory), or stroke (perfusion-diffusion equations). Each condition has associated mathematical frameworks that could guide machine learning toward mechanistic understanding.

Pharmacology: Pharmacokinetic-pharmacodynamic models could constrain learning of drug response patterns, enabling personalized dosing with fewer observations. Physics-based constraints would ensure predictions respect mass balance and physiological limits.

Biomechanics: Musculoskeletal models could guide rehabilitation planning and injury prevention. Constraints from Newton's laws and tissue mechanics would ensure biomechanically plausible movement predictions.

The framework demonstrates three principles for successful physics-informed medical AI:

1. Soft constraints are sufficient: Small physics loss weights ($\lambda_1 = 0.03$) provide substantial benefits without dominating empirical objectives

2. Interpretability enhances trust: Grounding predictions in established science enables clinical validation and adoption

3. Universal principles enable generalization: Physics-based features transfer across patients better than arbitrary learned patterns

### 4.6 Clinical translation pathway

Successful deployment requires addressing regulatory, technical, and implementation challenges beyond algorithm development:

Regulatory approval through FDA's Software as Medical Device pathway requires demonstrating safety and effectiveness through clinical trials. Our retrospective validation provides foundation for investigational device exemption, but prospective studies with predefined endpoints are essential. The interpretability features facilitate regulatory review by enabling examination of decision processes.

Technical infrastructure must support real-time processing of continuous EEG streams with redundancy for safety-critical applications. Cloud-based architectures could enable centralized processing with local edge devices for low-latency response. Integration with existing EEG systems through standard formats (EDF, MEF) and communication protocols (HL7 FHIR) is essential.

Clinical workflow integration requires careful design to minimize disruption while maximizing utility. User interfaces must present predictions, confidence levels, and explanations in clinically intuitive formats. Training programs for neurologists, nurses, and technicians should cover both technical operation and interpretation of physics-based parameters.

Economic evaluation should quantify cost-effectiveness through reduced hospitalizations, prevented injuries, and quality-adjusted life years. Our 35% reduction in monitoring requirements alone could save $30,000 per patient, with additional savings from prevented complications. Insurance coverage requires demonstrating both clinical efficacy and economic value.

Ethical considerations include managing false positives that cause unnecessary anxiety, ensuring equitable access across socioeconomic groups, and protecting patient privacy in cloud-based systems. Clear communication about prediction uncertainty and limitations is essential for informed consent.

## 5. Conclusion

We presented HP-GNN, a physics-informed graph neural network achieving robust epileptic seizure prediction through automated chimera state detection. By synergistically integrating Kuramoto oscillator dynamics with hypergraph architectures, our approach addresses three critical barriers that have prevented clinical deployment of seizure prediction systems: manual feature engineering, poor cross-patient generalization, and lack of interpretability.

Comprehensive evaluation on 22 pediatric patients (CHB-MIT) and 16 adults (IEEG.org) demonstrated substantial advances: 84.7% chimera detection accuracy (9.2% improvement over state-of-the-art), 89.3% seizure sensitivity with

90-minute prediction horizons, 79.8% cross-patient generalization (14.6% improvement), and 35% reduction in training data requirements. Clinical validation showed substantial agreement with neurologist assessments ($\kappa$=0.68) and biologically plausible learned parameters, confirming the model discovers genuine neurophysiological mechanisms rather than spurious correlations.

The mechanistic insights revealed through our analysis—including the three-phase seizure progression, spatial chimera organization, and quantitative synchronization parameters—advance fundamental understanding of seizure generation. The extended prediction horizons and acceptable false positive rates enable meaningful clinical interventions that could dramatically improve quality of life for millions with drug-resistant epilepsy.

Beyond epilepsy, this work establishes physics-informed geometric learning as a powerful paradigm for medical AI applications. The demonstrated benefits of incorporating domain knowledge— improved generalization, data efficiency, and interpretability—provide a blueprint for trustworthy AI systems that enhance rather than replace human expertise. Applications span cardiology, neurology, pharmacology, and biomechanics wherever mathematical models encode physiological principles.

Future research should pursue multicenter prospective validation, multimodal data integration, personalized model adaptation, and mechanistic refinements using sophisticated neural mass models. The ultimate goal remains deploying reliable, interpretable seizure prediction systems that transform epilepsy management while advancing the broader vision of physics-informed machine learning for healthcare.

In summary, HP-GNN represents a significant step toward clinically viable seizure prediction by bridging the gap between theoretical neuroscience and practical machine learning. The successful integration of chimera state theory with modern deep learning demonstrates that scientific knowledge and data-driven approaches are complementary tools for understanding and predicting complex biological phenomena. This work provides both a specific solution for epilepsy prediction and a general methodology for developing trustworthy medical AI systems grounded in scientific understanding.

## Supporting information

**S1 Fig. Training convergence analysis.** Four-panel figure showing: (a) total loss convergence across epochs for all cross-validation folds, (b) individual loss component trajectories, (c) gradient norm distribution during training, and (d) validation accuracy progression with early stopping markers.
(TIFF)

**S2 Fig. Electrode importance topographic maps.** GNNExplainer-generated importance scores displayed on standard 10–20 montage for representative patients from each dataset, with comparison to clinician-annotated seizure onset zones and inter-rater agreement statistics.
(TIFF)

**S1 Code. Complete implementation package.** Python source code including HP-GNN model architecture (PyTorch), training pipeline, data preprocessing scripts, evaluation metrics, and usage examples with documentation.
(ZIP)

## Acknowledgments

We thank the patients and families who contributed data to enable this research. The CHB-MIT database was provided by PhysioNet. Computational resources were provided by academic high- performance computing facilities.

This work is based upon research supported by Kermanshah University of Medical Science under project with Ethics ID "IR.KUMS.MED.REC.1404.253".

## Author contributions

**Conceptualization:** Masoud Amiri, Ershad Nedaei.

**Data curation:** Masoud Amiri.

**Formal analysis:** Masoud Amiri.

**Funding acquisition:** Masoud Amiri.

**Investigation:** Masoud Amiri.

**Methodology:** Masoud Amiri, Ershad Nedaei.

**Project administration:** Masoud Amiri, Bahador Makkiabadi.

**Resources:** Masoud Amiri.

**Software:** Masoud Amiri.

**Supervision:** Masoud Amiri, Bahador Makkiabadi.

**Validation:** Masoud Amiri, Bahador Makkiabadi.

**Visualization:** Masoud Amiri, Bahador Makkiabadi.

**Writing – original draft:** Masoud Amiri, Ershad Nedaei.

**Writing – review & editing:** Masoud Amiri, Ershad Nedaei, Bahador Makkiabadi.

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
