## [Decision Letter · Decision Letter 0]

13 Jan 2026

Dear Dr.  Amiri,

Thank you for submitting your manuscript to PLOS ONE. After careful consideration, we feel that it has merit but does not fully meet PLOS ONE’s publication criteria as it currently stands. Therefore, we invite you to submit a revised version of the manuscript that addresses the points raised during the review process.

We look forward to receiving your revised manuscript.

Kind regards,

Amirhossein Jafarian

Academic Editor

PLOS One

Journal Requirements:

2. Please ensure that your data sources are appropriately defined in the text, ensuring it is clear how access to the data may be acquired, and that the sources are appropriately cited (for examples, please see citation requirements specified at https://physionet.org/content/chbmit/1.0.0/).).).).

3. Thank you for uploading your study's underlying data set. Unfortunately, the repository you have noted in your Data Availability statement does not qualify as an acceptable data repository according to PLOS's standards.

At this time, please upload the minimal data set necessary to replicate your study's findings to a stable, public repository (such as figshare or Dryad) and provide us with the relevant URLs, DOIs, or accession numbers that may be used to access these data. For a list of recommended repositories and additional information on PLOS standards for data deposition, please see https://journals.plos.org/plosone/s/recommended-repositories....

4. Please include captions for your Supporting Information files at the end of your manuscript, and update any in-text citations to match accordingly. Please see our Supporting Information guidelines for more information: http://journals.plos.org/plosone/s/supporting-information....

5. We are unable to open your Supporting Information file “code.zip”. Please kindly revise as necessary and re-upload.

Reviewer's Responses to Questions

**Comments to the Author**

1. Is the manuscript technically sound, and do the data support the conclusions?

Reviewer #1: Partly

Reviewer #2: Yes

2. Has the statistical analysis been performed appropriately and rigorously?

Reviewer #1: Yes

Reviewer #2: Yes

3. Have the authors made all data underlying the findings in their manuscript fully available?

Reviewer #1: No

Reviewer #2: Yes

4. Is the manuscript presented in an intelligible fashion and written in standard English?

Reviewer #1: Yes

Reviewer #2: Yes

Reviewer #1: In this paper, the authors proposed a method based on the graph neural network to address epileptic seizure prediction. The manuscript clearly explained the clinical need for a reliable seizure prediction, emphasizing real-world constraints. The paper is well written; however, the research question should be introduced and explained more clearly, especially in the abstract.

• Section 2.1 (Dataset and preprocessing) needs significant improvement:

More details about the subjects and the data collection conditions should be provided.

Additionally, the details of preprocessing, such as the order of preprocessing stages, the number of ICA components, the selection criteria to identify noisy components, and the number of noisy components, must be reported to ensure the reproducibility of the method.

• The primary dataset (CHB-MIT) includes data from 22 patients, which is relatively small. This limits confidence in generalizability across age groups, seizure types, and clinical settings. The authors should extend the validation of the models using a larger dataset or an additional dataset.

• Since the proposed architecture is very complex and the training dataset is small, more discussion supported by quantitative analysis on computational cost and training stability would be necessary.

Reviewer #2: This paper proposes a hybrid graph neural network method for Epileptic Seizure Prediction using EEG. The physics-informed idea seems interesting to me and the obtained results support the effectiveness of the proposed model. There are however several concerns which I'd recommend the authors to address:

1. One major issue that I observed is the lack of proper comprehensive review of related works. It is expected the authors to include a separate section and thoroughly review the related work. Besides, majority of the references are outdated and very few recent works (e.g. 2023, 24, 25) are covered.

2. Presentation needs improvement; most images and graphs are of low quality; there are some grammatical and typos; too many short subsections exist, especially in section 1. The affiliations seem to be wrong. The authors should either extend them or merge some of them.

3. The novelty and advances to the state of the art should be clarified and quantified, as it is not very clear in the current version.

4. Some equations contain undefined variables like Eq. 2, 3.

5. It would be really beneficial to compare the proposed method with additional deep learning methods rather than GNN-based only. Also, compare the performance with ML techniques.

6. The authors emphasise on the explainability feature of the proposed method while very little (or no) discussion provided to address this in the results section.

.

Reviewer #1: No

Reviewer #2: No

---

## [Author Response · Author response to Decision Letter 1]

4 Feb 2026

Point-by-Point Response to Reviewers

Manuscript ID: PONE-D-25-65954

Physics-Informed Graph Neural Networks for Robust Cross-Patient Epileptic Seizure Prediction via Chimera State Detection

We sincerely thank the Academic Editor and both Reviewers for their constructive comments. We have carefully addressed each point with detailed responses and specific manuscript changes below.

JOURNAL REQUIREMENTS

Requirement 1: Style Requirements

Response:

We have reformatted the entire manuscript according to PLOS ONE style guidelines. All files have been renamed (Fig1.tif, Fig2.tif, Table.docx, etc.).

Requirement 2: Data Source Citations

Please ensure that your data sources are appropriately defined in the text, ensuring it is clear how access to the data may be acquired.

Response:

We have updated all dataset citations with proper DOIs and access information.

Requirement 3: Acceptable Data Repository

The repository you have noted in your Data Availability statement does not qualify as an acceptable data repository according to PLOS's standards. Please upload the minimal data set necessary to replicate your study's findings to a stable, public repository (such as figshare or Dryad).

Response:

We have uploaded all replication materials to Figshare (PLOS-approved repository):

• Complete HP-GNN source code

• Trained model weights (all folds)

• Preprocessing pipeline scripts

• Minimal replication dataset (5 patients)

• Evaluation scripts with expected outputs

Data Availability Statement

The datasets used in this study are available as follows:

1. CHB-MIT Scalp EEG Database: Publicly available through PhysioNet at https://physionet.org/content/chbmit/1.0.0/ under the Open Data Commons Open Database License v1.0.

2. SIENA Scalp EEG Database: Publicly available through PhysioNet at https://physionet.org/content/siena-scalp-eeg/1.0.0/.

3. IEEG.org Database: Available upon institutional Data Use Agreement at https://www.ieeg.org.

4. Code and Replication Materials: All source code, trained model weights, preprocessing scripts, evaluation code, and a minimal replication dataset are available with supplementary material.

Requirement 4: Supporting Information Captions

Please include captions for your Supporting Information files at the end of your manuscript.

Response:

We have added comprehensive captions for all Supporting Information files.

SUPPORTING INFORMATION:

SUPPORTING INFORMATION

S1 Fig. Training convergence analysis. Four-panel figure showing: (a) total loss convergence across epochs for all cross-validation folds, (b) individual loss component trajectories, (c) gradient norm distribution during training, and (d) validation accuracy progression with early stopping markers.

S2 Fig. Electrode importance topographic maps. GNNExplainer-generated importance scores displayed on standard 10-20 montage for representative patients from each dataset, with comparison to clinician-annotated seizure onset zones and inter-rater agreement statistics.

S1 Code. Complete implementation package. Python source code including HP-GNN model architecture (PyTorch), training pipeline, data preprocessing scripts, evaluation metrics, and usage examples with documentation.

Requirement 5: ZIP File Format

We are unable to open your Supporting Information file 'code.zip'. Please kindly revise and re-upload.

Response:

We apologize for the technical issue. The file has been recompressed using standard ZIP format without password protection. We verified successful extraction on Windows 10/11, macOS Ventura, and Ubuntu 22.04. Large model files have been split into smaller archives (<25MB). A comprehensive README.txt is included.

REVIEWER #1 COMMENTS

Comment 1.1: Research Question in Abstract

The paper is well written; however, the research question should be introduced and explained more clearly, especially in the abstract.

Response:

We thank the reviewer for this feedback. We have revised the abstract to explicitly and prominently state the research question.

Research Question

Research Question: Can integrating physics-based constraints from Kuramoto oscillator theory with graph neural networks enable automated, robust, and interpretable chimera-based seizure prediction that generalizes across patients?

Comment 1.2: Dataset and Preprocessing Details

Section 2.1 (Dataset and preprocessing) needs significant improvement: More details about the subjects and the data collection conditions should be provided. Additionally, the details of preprocessing, such as the order of preprocessing stages, the number of ICA components, the selection criteria to identify noisy components, and the number of noisy components, must be reported.

Response:

We have substantially expanded Section 2.1 with all requested details.

Subject Details (Section 2.1):

Subject Demographics and Data Collection

Table 2 summarizes the demographic characteristics across all three datasets. The CHB-MIT database comprises pediatric patients (age range 1.5-22 years, mean 11.3 ± 5.2 years) recorded at Boston Children's Hospital using a standard clinical EEG system. The IEEG.org dataset includes adult patients (age range 18-58 years, mean 34.6 ± 12.8 years) with intracranial electrode implants for pre-surgical evaluation. The SIENA database contains adult patients (age range 21-65 years, mean 41.2 ± 15.3 years) recorded at the University of Siena, Italy, using a Micromed SystemPlus acquisition system with a sampling rate of 512 Hz and 29 channels in the extended 10-20 system.

Table 2 summarizes the demographic characteristics across all three datasets.

Dataset N Age (years) Sex (F/M) Seizure Types Recording Hours

CHB-MIT 22 1.5-22 (11.3±5.2) 17/5 Focal, Generalized 844

IEEG.org 16 18-58 (34.6±12.8) 9/7 Focal (intracranial) 312

SIENA 14 21-65 (41.2±15.3) 8/6 Focal aware/impaired 186

The diversity in age groups, seizure types, and recording systems across datasets enables comprehensive evaluation of cross-patient generalization.

Preprocessing Pipeline (Section 2.1):

Preprocessing Pipeline

All EEG data underwent a standardized preprocessing pipeline implemented in Python using MNE-Python. The preprocessing stages were applied in the following order:

Step 1 - Resampling: All recordings were resampled to a uniform 256 Hz using scipy.signal.resample_poly with a Hamming window to prevent aliasing artifacts.

Step 2 - Bandpass Filtering: A 4th-order Butterworth bandpass filter (1-50 Hz) was applied using zero-phase filtering (scipy.signal.filtfilt) to remove DC drift and high-frequency noise while preserving physiologically relevant frequency bands.

Step 3 - Independent Component Analysis (ICA): FastICA decomposition was performed with 20 components (random seed=42, tolerance=10⁻⁴, maximum iterations=200). Artifactual components were identified using five automated criteria: (1) kurtosis > 5.0 indicating eye blinks or muscle artifacts, (2) high-frequency power ratio > 40% (power above 30 Hz relative to total) indicating EMG contamination, (3) frontal topography with > 70% weight on Fp1/Fp2 electrodes indicating ocular artifacts, (4) low temporal autocorrelation < 0.3 at 1-second lag indicating random noise, and (5) correlation > 0.8 with EOG reference channels when available. Components meeting any criterion were flagged for removal. On average, 3.2 ± 1.1 components per patient (range: 1-6) were removed.

Step 4 - Normalization: Channel-wise z-score normalization was applied using statistics computed exclusively from interictal periods to prevent information leakage from ictal segments.

Comment 1.3: Dataset Size and Generalizability

The primary dataset (CHB-MIT) includes data from 22 patients, which is relatively small. This limits confidence in generalizability across age groups, seizure types, and clinical settings. The authors should extend the validation using a larger dataset or an additional dataset.

Response:

We have added validation on a third dataset (SIENA) and performed comprehensive subgroup analyses.

SIENA Dataset (Section 2.1):

SIENA Scalp EEG Database

To address concerns about generalizability across adult populations and different clinical settings, we performed additional validation on the SIENA Scalp EEG Database [43]. This dataset contains recordings from 14 adult patients (mean age 41.2 ± 15.3 years, range 21-65 years, 8 female) with medically refractory focal epilepsy. Recordings were acquired at the Neurology and Clinical Neurophysiology Unit, University of Siena, Italy, using a Micromed SystemPlus system with 29 channels in the extended 10-20 configuration at 512 Hz sampling rate. This dataset provides independent validation across: (1) different geographical regions (Europe vs. North America), (2) different acquisition systems (Micromed vs. clinical EEG), (3) different patient demographics (European adult vs. North American pediatric), and (4) different annotation protocols.

Table Cross-Dataset Performance – Added to Section 3.1 (Results):

Table 7 demonstrates consistent performance across all three datasets, with modest variation (5.5% range) substantially smaller than the 20-35% degradation typical of purely data-driven approaches.'

Dataset Accuracy 95% CI Sensitivity FP/hr

CHB-MIT 84.7% 82.3-87.1% 89.3% 0.48

IEEG.org 79.2% 75.8-82.6% 83.1% 0.62

SIENA 82.3% 79.1-85.5% 86.7% 0.54

The robust performance across diverse patient populations, acquisition systems, and clinical settings supports the generalizability of our approach.

References (2023-2025):

• Detti P, de Lara GZM, Bruni R, et al. Siena Scalp EEG Database (version 1.0.0). PhysioNet. 2023. https://doi.org/10.13026/5d4a-j060

Comment 1.4: Computational Cost and Training Stability

Since the proposed architecture is very complex and the training dataset is small, more discussion supported by quantitative analysis on computational cost and training stability would be necessary.

Response:

We have added a comprehensive new subsection analyzing computational requirements and training stability.

Section 2.6.1 (new subsection):

2.6.1 Computational Analysis and Training Stability

Given the architectural complexity of HP-GNN and the relatively limited training data, we conducted comprehensive analysis of computational requirements and training stability to ensure reproducibility and practical deployment feasibility.

Computational Resources

All experiments were conducted on a workstation equipped with an NVIDIA A100 GPU (40GB VRAM), AMD EPYC 7742 CPU (64 cores), and 256GB RAM. Table 3 summarizes the computational requirements for HP-GNN compared to baseline methods.

Computational Comparison:

Table 3 summarizes the computational requirements for HP-GNN compared to baseline methods.'

Method Parameters Training (h/fold) Inference (ms) GPU Memory

HP-GNN (Ours) 2.47M 4.2 ± 0.3 12.3 ± 1.2 8.4 GB

Transformer 8.92M 6.1 ± 0.4 45.2 ± 3.1 12.8 GB

ResNet-18 11.2M 3.5 ± 0.2 15.4 ± 1.4 6.2 GB

BiLSTM 3.14M 2.8 ± 0.2 8.7 ± 0.8 4.1 GB

TCN 4.21M 3.1 ± 0.2 14.1 ± 1.2 5.3 GB

HP-GNN achieves real-time inference capability (>80 windows/second at 256 Hz), enabling practical deployment in clinical monitoring systems.

Section 2.6.1:

Training Stability Analysis

To ensure reliable convergence despite the complex architecture, we monitored multiple stability metrics throughout training:

Loss Convergence: The total loss decreased monotonically from 2.34 ± 0.12 (epoch 1) to 0.42 ± 0.08 (epoch 100) across all cross-validation folds, with no oscillations or divergence observed. Individual loss components showed balanced optimization: chimera detection loss (1.02 → 0.18), state classification loss (0.87 → 0.15), time regression loss (0.32 → 0.06), and physics regularization loss (0.11 → 0.02).

Gradient Flow: Mean gradient magnitudes remained within the stable range [0.001, 0.1] throughout training, with standard deviation < 0.02. Gradient clipping (max norm = 1.0) was triggered in only 2.3% of training batches, predominantly during the first 10 epochs.

Early Stopping: Validation accuracy convergence occurred at epoch 87.3 ± 8.4 across folds, well before the 100-epoch maximum, indicating that the model learned meaningful representations without overfitting.

Generalization Gap: The difference between training accuracy (92.1%) and validation accuracy (84.7%) was 7.4%, substantially smaller than the 15-25% gap typically observed in deep learning models trained on similar dataset sizes, suggesting that physics constraints provide effective regularization.

Cross-Fold Consistency: Coefficient of variation (CV) was below 10% for all performance metrics across the four cross-validation folds, demonstrating robust and reproducible results.

Detailed training curves are provided in Supplementary Figure S1.

REVIEWER #2 COMMENTS

Comment 2.1: Comprehensive Literature Review

One major issue that I observed is the lack of proper comprehensive review of related works. It is expected the authors to include a separate section and thoroughly review the related work. Besides, majority of the references are outdated and very few recent works (e.g. 2023, 24, 25) are covered.

Response:

We sincerely thank the reviewer for this critical feedback. We have added a dedicated Section 1.5 'Related Work and Positioning' with comprehensive coverage of recent literature (14 new references from 2023-2025).

Section 1.5 (Related Work):

1.5 Related Work and Positioning

We organize the related work into five thematic areas, emphasizing recent advances (2023-2025) that contextualize our contributions.

Deep Learning for Seizure Prediction

Recent years have witnessed significant advances in deep learning approaches for seizure prediction. Zhang et al. [27] developed a large-scale self-supervised pre-training framework using over 10,000 hours of EEG data, achieving 87.2% sensitivity on the CHB-MIT dataset. However, their approach showed 25% accuracy degradation when applied to new patients without fine-tuning, highlighting persistent generalization challenges. Song et al. [28] introduced the EEG Conformer architecture combining convolutional and transformer layers, achieving state-of-the-art results on multiple EEG classification benchmarks. Pinto et al. [29] provided a comprehensive review of deep learning architectures for seizure prediction, identifying cross-patient generalization and clinical interpretability as the two most critical remaining challenges. Tang et al. [30] proposed self-supervised graph neural networks that improved data efficiency by 40% through contrastive pre-training, though their approach remained limited to patient-specific models.

Graph Neural Networks for Neurophysiological Signals

Graph-based approaches have gained prominence for modeling brain connectivity. Cui et al. [31] introduced BrainNetFormer, applying graph transformers to brain networks for seizure prediction with attention-based interpretability. Liu et al. [32] developed temporal graph neural networks that explicitly model evolving connectivity patterns, achieving 81.3% accuracy on CHB-MIT with improved temporal resolution. Chen et al. [33] proposed multi-scale graph convolutional networks capturing both local and global connectivity patterns. Wang et al. [34] combined spatial attention mechanisms with temporal graph convolutions, demonstrating that explicit connectivity modeling outperforms treating EEG channels as independent time series. However, none of these approaches incorporated physics-based constraints, limiting their ability to leverage domain knowledge about neural dynamics.

Physics-Informed Machine Learning for Neuroscience

Physics-informed approaches have shown promise for scientific machine learning. Kumar et al. [35] extended this framework to neural oscillator models, showing that Kuramoto-inspired constraints improved seizure detection accuracy by 8% while reducing training data requirements by 30%. However, their approach used simplified pairwise interactions rather than higher-order hypergraph structures. Our work advances this direction by integrating Kuramoto dynamics with hypergraph neural networks, enabling capture of collective synchronizatio

---

## [Decision Letter · Decision Letter 1]

5 Mar 2026

Physics-Informed Graph Neural Networks for Robust Cross-Patient Epileptic Seizure Prediction via Chimera State Detection

PONE-D-25-65954R1

Dear Dr. Amiri

We’re pleased to inform you that your manuscript has been judged scientifically suitable for publication and will be formally accepted for publication once it meets all outstanding technical requirements.

Kind regards,

Amirhossein Jafarian

Academic Editor

PLOS One

Additional Editor Comments (optional):

Reviewers' comments:

Reviewer's Responses to Questions

**Comments to the Author**

Reviewer #2: All comments have been addressed

2. Is the manuscript technically sound, and do the data support the conclusions?

Reviewer #2: Yes

3. Has the statistical analysis been performed appropriately and rigorously?

Reviewer #2: Yes

4. Have the authors made all data underlying the findings in their manuscript fully available?

Reviewer #2: (No Response)

5. Is the manuscript presented in an intelligible fashion and written in standard English?

Reviewer #2: (No Response)

Reviewer #2: the authors have made good effort to address my concerns, and I am happy to recommend this paper for publication.

.

Reviewer #2: No

---

## [Editor Report · Acceptance letter]

PONE-D-25-65954R1

PLOS One

Dear Dr. Amiri,

I'm pleased to inform you that your manuscript has been deemed suitable for publication in PLOS One. Congratulations! Your manuscript is now being handed over to our production team.

Kind regards,

on behalf of

Dr. Amirhossein Jafarian

Academic Editor

PLOS One